# Boosting with Multiple Sources

**Corinna Cortes**
Google Research
New York, NY 10011
corinna@google.com

**Mehryar Mohri**
Google & Courant Institute
New York, NY 10012
mohri@google.com

**Dmitry Storcheus**
Courant Institute & Google
New York, NY 10012
dstorcheus@google.com

**Ananda Theertha Suresh**
Google Research
New York, NY 10011
theertha@google.com

## Abstract

We study the problem of learning accurate ensemble predictors, in particular boosting, in the presence of multiple source domains. We show that the standard convex combination ensembles in general cannot succeed in this scenario and adopt instead a domain-weighted combination. We introduce and analyze a new boosting algorithm, MULTIBOOST, for this scenario and show that it benefits from favorable theoretical guarantees. We also report the results of several experiments with our algorithm demonstrating that it outperforms natural baselines on multi-source text-based, image-based and tabular data. We further present an extension of our algorithm to the federated learning scenario and report favorable experimental results for that setting as well. Additionally, we describe in detail an extension of our algorithm to the multi-class setting, MCMULTIBOOST, for which we also report experimental results.

## 1 Motivation

Ensemble methods such as Bagging, AdaBoost, Stacking, error-correction techniques, Bayesian averaging, AdaNet or other adaptive methods for learning neural networks are general machine learning techniques used to combine several predictors to devise a more accurate one (Breiman, 1996; Freund and Schapire, 1997; Smyth and Wolpert, 1999; MacKay, 1991; Freund et al., 2004; Cortes et al., 2014; Kuznetsov et al., 2014; Cortes et al., 2017). These techniques are often very effective in practice and benefit from favorable margin-based learning guarantees (Schapire et al., 1997). These algorithms assume access to a training sample drawn from the target distribution. But, in many applications, the learner receives labeled data from multiple source domains that it seeks to use to find a good predictor for target domains, which are typically assumed to be mixtures of source distributions (Mansour et al., 2008, 2009a; Hoffman et al., 2018, 2021; Cortes et al., 2021; Muandet et al., 2013; Xu et al., 2014; Hoffman et al., 2012; Saito et al., 2019; Wang et al., 2019a). How can we generalize ensemble methods such as boosting to this scenario?

Several related problems have been tackled in previous work. In the special case of a single target domain, boosting solutions have been derived (Dai et al., 2007; Yao and Doretto, 2010). These algorithms have been further improved by boosting base predictors jointly with source features (or feature *views*) that are predictive in the target (Yuan et al., 2017; Cheng et al., 2013; Zhang et al., 2014; Xu and Sun, 2012, 2011). Such methods have been widely adopted in various domain adaptation scenarios for different types of data. For example, Huang et al. (2010, 2012) showed that by selecting a base learner jointly with a feature that is predictive across multiple domains at every boosting step, one can achieve higher accuracy than standard transfer learning methods. Moreover, the margin

35th Conference on Neural Information Processing Systems (NeurIPS 2021).

provided by boosting-style algorithms can aid in transfer learning where target domain is unlabelled. Habrard et al. (2013) have developed an algorithm that jointly minimizes the source domain error and margin violation proportion on the target domain. Wang et al. (2019a) have demonstrated that boosting classifiers from different domains can be carried out online. Taherkhani et al. (2020) and Becker et al. (2013) have shown that multi-source boosting can be combined with Deep Neural Networks for multi-task learning on large scale datasets.

We give a more detailed discussion of this related prior work in Appendix A. However, as we demonstrate in this paper, several critical questions related to the presence of multiple domains have not been fully addressed by existing boosting methods from both the algorithmic and the theoretical point of view. First, what should be the form of the ensemble solutions? We show that, in general, the standard convex combinations used in much of previous work may not succeed in this problem (Section 2). Instead, we put forward Q-ensembles, which are convex combinations weighted by a domain classifier Q, that is, $Q(k|x)$ is the conditional probability of domain $k$ given input point $x$. This is inspired by similar Q-ensembles (Cortes, Mohri, Suresh, and Zhang, 2021) or distribution-weighted combinations (Mansour, Mohri, and Rostamizadeh, 2008, 2009a; Hoffman, Mohri, and Zhang, 2018, 2021) used in the context of multiple-source adaptation and crucially differentiates our work from that of past studies of ensemble methods. Our learning scenario strictly generalizes previous algorithms. In the special case of a single domain, the form of our ensemble solutions coincides with that of the familiar ensembles used in AdaBoost. In Appendix J, we further compare our results and guarantees with the multiple-source adaptation algorithms of (Hoffman, Mohri, and Zhang, 2018, 2021; Cortes, Mohri, Suresh, and Zhang, 2021), when using AdaBoost to derive a predictor for each domain.

Second, what should be the form of the objective? Unlike existing boosting methods which optimize for a specific target distribution or domain, we seek a solution that is accurate for any mixture of the source distributions, where the mixture weights may be constrained to be in a subset of the simplex. This conforms with the learning goal in this scenario where the target distribution is typically assumed to be a mixture of the source ones and is further inspired by the formulation of the *agnostic loss* in the related context of federated learning (Mohri, Sivek, and Suresh, 2019). Additionally, this formulation can be viewed as more risk-averse and robust, and it also ensures more favorable fairness properties, since it does not favor any distribution possibly biased towards some subset of the source domains. In Section 3, we introduce and analyze an algorithm, MULTIBOOST, that is based on such an objective. This further distinguishes our boosting algorithm from related prior work. In Appendix L, we further give a detailed description of a multi-class extension of our algorithm, MCMULTIBOOST, including its pseudocode, the discussion of some of its properties, and the results of several experiments.

Third, in Section 4, we present a theoretical analysis of our MULTIBOOST algorithm, including margin-based generalization bounds that hold for any target mixture of the source distributions. In Appendix F, we derive finer and more flexible learning bounds that can provide more favorable guarantees, in particular when the family of target mixtures is more constrained.

Fourth, in Section 5, we describe FEDMULTIBOOST, an extension of our MULTIBOOST algorithm adapted to the distributed learning scenario of *federated learning*, which is increasingly important in a wide array of applications (Kairouz et al., 2021), including healthcare (Brisimi et al., 2018). FEDMULTIBOOST allows the server to train ensemble models that benefit from clients' data, while the data remains with clients. We provide a detailed description of our algorithm, as well as a brief analysis of the communication costs, which are critical in this scenario. Appendix G contains experimental results comparing FEDMULTIBOOST with several baselines.

Finally, in Section 6, we report the results of extensive experiments with our MULTIBOOST algorithm. We start with a detailed description of the learning scenario we consider.

## 2  Learning scenario

Let $\mathcal{X}$ denote the input space and $\mathcal{Y} = \{-1, +1\}$ the output space associated to binary classification. We consider a scenario where the learner receives labeled samples from $p$ source domains, each defined by a distribution $\mathcal{D}_k$ over $\mathcal{X} \times \mathcal{Y}$, $k \in [p]$. We denote by $S_k = ((x_{k,1}, y_{k,1}), \ldots, (x_{k,m_k}, y_{k,m_k})) \in (\mathcal{X} \times \mathcal{Y})^{m_k}$ the labeled sample of size $m_k$ received from source $k$, which is drawn i.i.d. from $\mathcal{D}_k$. For any function $f \colon \mathcal{X} \to \mathbb{R}$ and distribution $\mathcal{D}$ over $\mathcal{X} \times \mathcal{Y}$, let

$\mathcal{L}(\mathcal{D}, f)$ be the expected loss of $f$, that is $\mathcal{L}(\mathcal{D}, f) = \mathbb{E}_{(x,y)\sim\mathcal{D}}[\ell(f(x), y)]$, where $\ell$ is the binary loss $\ell(f(x), y) = \mathbb{I}(yf(x) \leq 0)$.

For any $k \in [p]$, let $\mathcal{H}_k$ denote a hypothesis set of functions mapping from $\mathcal{X}$ to $[-1, +1]$, $|\mathcal{H}_k| = N_k$. The objective of the learner is to find a predictor $f$ that is accurate for *any* target distribution $\mathcal{D}_\lambda$ that is a mixture of the source distributions, where $\lambda$ may be in a subset $\Lambda$ of the simplex. Thus, $\mathcal{D}_\lambda$ can written as $\mathcal{D}_\lambda = \sum_{k=1}^{p} \lambda_k \mathcal{D}_k$, with $\lambda = (\lambda_1, \ldots, \lambda_p) \geq 0$ and $\sum_{k=1}^{p} \lambda_k = 1$. This leads to the following *agnostic loss* of a predictor $f$ (Mohri et al., 2019):

$$\mathcal{L}(\mathcal{D}_\Lambda, f) = \max_{\lambda \in \Lambda} \mathcal{L}(\mathcal{D}_\lambda, f). \tag{1}$$

To come up with a predictor $f$, the learner seeks an ensemble of functions from the base classes $\mathcal{H}_k$. What should be the form of such ensembles? A natural solution is a convex combinations of the form

$$f = \sum_{l=1}^{p} \sum_{j=1}^{N_l} \alpha_{l,j} h_{l,j}, \tag{2}$$

where $h_{l,j} \in \mathcal{H}_l$ and $\alpha_{l,j} \geq 0$, $\sum_{l=1}^{p} \sum_{j=1}^{N_l} \alpha_{l,j} = 1$, as in prominent algorithms such as BAGGING (Breiman, 1999) or ADABOOST (Freund and Schapire, 1997). It is not hard to show, however, that for some distributions, any such convex combination of base predictors would lead to a poor solution.

**Proposition 1.** *There exist distributions $\mathcal{D}_1$ and $\mathcal{D}_2$ and hypotheses $h_1$ and $h_2$ with $\mathcal{L}(\mathcal{D}_1, h_1) = 0$ and $\mathcal{L}(\mathcal{D}_2, h_2) = 0$ such that $\mathcal{L}\left(\frac{1}{2}(\mathcal{D}_1 + \mathcal{D}_2), \alpha h_1 + (1-\alpha)h_2\right) \geq \frac{1}{2}$ for any $\alpha \in [0, 1]$,*

This result is similar to (Mansour et al., 2008, Theorem 1) or similar statements in (Mansour et al., 2008, 2009a; Hoffman et al., 2018, 2021; Cortes et al., 2021), however, there are some technical differences. In particular, we are considering here a binary classification loss and not the absolute loss. The proof is given in Appendix B.

Note that in the example of the proposition, the base predictors $h_1, h_2$ are both perfect for their respective domains. Nevertheless, unlike the common case of a single source domain, standard convex combinations in general may perform poorly. Thus, we will consider instead the following form for the ensembles of base predictors:

$$\forall x \in \mathcal{X}, \quad f(x) = \sum_{l=1}^{p} \sum_{j=1}^{N_l} \alpha_{l,j} \mathsf{Q}(l|x) h_{l,j}(x), \tag{3}$$

where $\mathsf{Q}(l|x)$ denotes the conditional probability of domain $l$ given $x$. This helps us account for the fact that the learner combines base predictors from different domains. For a given point $x$, experts from the domains where $x$ is more likely to appear are allocated more weight in the voted combination. The following result further substantiates the hope for this method to succeed.

**Proposition 2.** *For the same distributions $\mathcal{D}_1$ and $\mathcal{D}_2$ and hypotheses $h_1$ and $h_2$ as in Proposition 1, the equality $\mathcal{L}(\mathcal{D}_\lambda, (\alpha \mathsf{Q}(1|\cdot)h_1 + (1-\alpha)\mathsf{Q}(2|\cdot)h_2)) = 0$ holds for any $\lambda \in \Delta$ and any $\alpha \in (0, 1)$.*

The proof is also given in Appendix B. Thus, our Q-ensembles can be substantially more effective in the multiple-source adaptation problem considered. Note that, in the special case of a single domain, the Q-combinations coincide with standard convex combinations, since $\mathsf{Q}(k|x) = 1$ for all $x$.

As in the standard boosting scenario, we will adopt a weak-learning assumption. However, our assumption here must hold for each source domain: for each domain $k \in [p]$ and any distribution D over $S_k$, there exists a base classifier $h \in \mathcal{H}_k$ such that the weighted loss of $h$ is $\gamma$-better than random: $\frac{1}{2}[1 - \mathbb{E}_{i\sim\mathsf{D}}[y_{k,i}h(x_{k,i})]] \leq \frac{1}{2} - \gamma$, for some edge value $\gamma > 0$. This is equivalent to the existence of a weak-learning algorithm for each domain, which is a mild assumption. As in the standard boosting scenario, this corresponds to the existence of a a good *rule of thumb* for each domain. Unlike the standard scenario, however, here, we seek a Q-ensemble and further require that ensemble to be accurate for any target mixture $\mathcal{D}_\lambda, \lambda \in \Lambda$. Note that the definition of the hypothesis set $\mathcal{H}_k$ and the weak-learning assumption for domain $k$ are intimately related: the elements of $\mathcal{H}_k$ are typically simple predictors sufficient to guarantee the weak-learning assumption for that domain.

In the next section, we present an algorithm, MULTIBOOST, for deriving an accurate Q-ensemble for any target mixture domain that belongs to the convex combination of the source domains. In Appendix L, we further present a multi-class extension of our algorithm, MCMULTIBOOST.

## 3 Algorithm

Let $\Phi$ be a convex, increasing and differentiable function such that $u \mapsto \Phi(-u)$ upper-bounds the binary loss $u \mapsto 1_{u \leq 0}$. $\Phi$ could be the exponential function as in ADABOOST or the logistic function, as in logistic regression. Using $\Phi$ to upper-bound the agnostic loss leads to the following objective function for an ensemble $f$ defined by (3) for any $\alpha = (\alpha_{l,j})_{(l,j) \in [p] \times [N_l]} \geq 0$:

$$F(\alpha) = \max_{\lambda \in \Lambda} \sum_{k=1}^{p} \frac{\lambda_k}{m_k} \sum_{i=1}^{m_k} \Phi\left(-y_{k,i} \sum_{l=1}^{p} \sum_{j=1}^{N_l} \alpha_{l,j} \mathsf{Q}(l|x_{k,i}) h_{l,j}(x_{k,i})\right). \tag{4}$$

Since a convex function composed with an affine function is convex and a sum of convex functions is convex, $F$ is convex as the maximum of a set of convex functions. In this section, we will consider the case where the set $\Lambda$ coincides with the full simplex, that is $\Lambda = \Delta$. $F$ can then be expressed more simply as $F = \max_{k=1}^{p} F_k$, with $F_k(\alpha) = \frac{1}{m_k} \sum_{i=1}^{m_k} \Phi\left(-y_{k,i} \sum_{l=1}^{p} \sum_{j=1}^{N_l} \alpha_{l,j} \mathsf{Q}(l|x_{k,i}) h_{l,j}(x_{k,i})\right)$.

It is known that ADABOOST coincides with coordinate descent applied to an exponential loss objective function (Duffy and Helmbold, 1999; Schapire and Freund, 2012; Mohri et al., 2018). Our algorithm similarly applies coordinate descent to the objective just described, as with other boosting-type algorithms (Mason et al., 1999; Cortes et al., 2014; Kuznetsov et al., 2014). While convex, $F$ is not differentiable and, in general, coordinate descent may not succeed in such cases (Tseng, 2001; Luo and Tseng, 1992). This is because the algorithm may be *stuck* at a point where no progress is possible along any of the axis directions, while there exists a favorable descent along some other direction. However, we will show that, under the weak-learning assumption we adopted, at any point $\alpha$ and for each active function $F_k$, that is $F_k$ such that $F_k(\alpha) = F(\alpha)$, there exists a coordinate direction along which a descent progress is possible. We will assume that these directions are also descent directions for $F$. More generally, it suffices in fact that one such direction admits this guarantee. Alternatively, one can replace the maximum with a *soft-max*, that is the $(x_1, \ldots, x_k) \mapsto \frac{1}{\mu} \log(\sum_{k=1}^{p} e^{\mu x_k})$ for $\mu > 0$, which leads to a continuously differentiable function with a close approximation for $\mu$ sufficiently large.

**Description**. Let $\alpha_{t-1}$ denote the value of the parameter vector $\alpha = (\alpha_{l,j})$ at the end of the $(t-1)$th iteration and $f_{t-1}$ the corresponding function: $f_{t-1} = \sum_{l=1}^{p} \sum_{j=1}^{N_l} \alpha_{t-1,l,j} \mathsf{Q}(l|\cdot) h_{l,j}$. Coordinate descent at iteration $t$ consists of choosing a direction $\mathbf{e}_{q,r}$ corresponding to base classifier $h_{q,r}$ and a step value $\eta > 0$ to minimize $F(\alpha_{t-1} + \eta \mathbf{e}_{q,r})$.

To select a direction, we consider the subdifferential of $F$ along any $\mathbf{e}_{q,r}$. Since functions $F_k$ are differentiable, by the subdifferential calculus for the maximum of functions, the subdifferential of $F$ at $\alpha_{t-1}$ along the direction $\mathbf{e}_{q,r}$ is given by:

$$\partial F(\alpha_{t-1}, \mathbf{e}_{q,r}) = \operatorname{conv}\{F_k'(\alpha_{t-1}, \mathbf{e}_{q,r}) \colon k \in \mathcal{K}_t\},$$

where $F_k'(\alpha_{t-1}, \mathbf{e}_{q,r})$ is the directional derivative of $F_k$ at $\alpha_{t-1}$ along the direction $\mathbf{e}_{q,r}$ and where $\mathcal{K}_t = \{k \in [p] \colon F_k(\alpha_{t-1}) = F(\alpha_{t-1})\}$. We will therefore consider the direction $\mathbf{e}_{q,r}$ with the largest absolute directional derivative value $|F_k'(\alpha_{t-1}, \mathbf{e}_{q,r})|$, $k \in \mathcal{K}_t$, but will restrict ourselves to $q = k$ since, as we shall see, that will suffice to guarantee a non-zero directional gradient. To do so, we will express $F_k'(\alpha_{t-1}, \mathbf{e}_{k,r})$ in terms of the distribution $\mathsf{D}_t(k, \cdot)$ over $S_k$ defined by $\mathsf{D}_t(k, i) = \frac{\Phi'(-y_{k,i} f_{t-1}(x_{k,i}))}{Z_{t,k}}$, with $Z_{t,k} = \sum_{i=1}^{m_k} \Phi'(-y_{k,i} f_{t-1}(x_{k,i}))$, for all $i \in [m_k]$:

$$F_k'(\alpha_{t-1}, h_{k,r}) = \frac{1}{m_k} \sum_{i=1}^{m_k} -y_{k,i} \mathsf{Q}(k|x_{k,i}) h_{k,r}(x_{k,i}) \Phi'\left(-y_{k,i} f_{t-1}(x_{k,i})\right) = \frac{Z_{t,k}}{m_k} [2\epsilon_{t,k,r} - 1],$$

where $\epsilon_{t,k,r} = \frac{1}{2}\left[1 - \mathbb{E}_{i \sim \mathsf{D}_t(k,\cdot)}[y_{k,i} \mathsf{Q}(k|x_{k,i}) h_{k,r}(x_{k,i})]\right]$ denotes the *weighted error* of $\mathsf{Q}(k|\cdot) h_{k,r}$. For any $s \in [m_k]$, since $x_{k,s}$ is a sample drawn from $\mathcal{D}_k$, we have $\mathsf{Q}(k|x_{k,s}) > 0$ and therefore we have: $\overline{\mathsf{Q}}_{t,k} = \sum_{s=1}^{m_k} \mathsf{D}_t(k,s) \mathsf{Q}(k|x_{k,s}) > 0$. Thus, we can write $\mathbb{E}_{i \sim \mathsf{D}_t(k,\cdot)}[y_{k,i} \mathsf{Q}(k|x_{k,i}) h_{k,r}(x_{k,i})]$ as

$$\sum_{i=1}^{m_k} \frac{\mathsf{D}_t(k,i) \mathsf{Q}(k|x_{k,i}) y_{k,i} h_{k,r}(x_{k,i})}{\overline{\mathsf{Q}}_{t,k}} \overline{\mathsf{Q}}_{t,k}, = \mathbb{E}_{i \sim \mathsf{D}_t'(k,\cdot)}[y_{k,i} h_{k,r}(x_{k,i})] \overline{\mathsf{Q}}_{t,k},$$

where $\mathsf{D}_t'(k,i) = \frac{\mathsf{D}_t(k,i) \mathsf{Q}(k|x_{k,i})}{\overline{\mathsf{Q}}_{t,k}}$. By our weak-learning assumption, there exists $r \in N_k$ such that $\mathbb{E}_{i \sim \mathsf{D}_t'(k,\cdot)}[y_{k,i} h_{k,r}(x_{k,i})] \geq \gamma > 0$. For that choice of $r$, we have $\epsilon_{t,k,r} < \frac{1}{2} - \overline{\gamma}$, with $\overline{\gamma} = \gamma \overline{\mathsf{Q}}_{t,k} > 0$.

In view of that, it suffices for us to search along the directions $h_{k,r}$ and we do not need to consider the directional derivative of $F_k$ along directions $h_{q,r}$ with $q \neq k$. The direction chosen by our coordinate descent algorithm is thus defined by: $\operatorname{argmax}_{k \in \mathcal{K}_t, r \in [N_k]} \frac{Z_{t,k}}{m_k}[1 - 2\epsilon_{t,k,r}]$. Given the direction $\mathbf{e}_{k,r}$, the optimal step value $\eta$ is $\operatorname{argmin}_{\eta > 0} F(\alpha_{t-1} + \eta \mathbf{e}_{k,r})$. The pseudocode of our algorithm, MULTIBOOST, is provided in Figure 1. In the most general case, $\eta$ can be found via a line search or other numerical methods.

**Step size**. In some special cases, the line search can be executed using a simpler expression by using an upper bound on $F(\alpha_{t-1} + \eta \mathbf{e}_{k,r})$. Using the convexity of $\Phi$, since $y_{l,i} \mathsf{Q}(k|x_{l,i}) h_{k,r}(x_{l,i}) = \frac{1 + y_{l,i} \mathsf{Q}(k|x_{l,i}) h_{k,r}(x_{l,i})}{2} \cdot (+1) + \frac{1 - y_{l,i} \mathsf{Q}(k|x_{l,i}) h_{k,r}(x_{l,i})}{2} \cdot (-1)$, the following holds for all $\eta \in \mathbb{R}$:

$$\Phi(-y_{l,i} f_{t-1}(x_{l,i}) - \eta\, y_{l,i} \mathsf{Q}(k|x_{l,i}) h_{k,r}(x_{l,i})) \leq \frac{1 + y_{l,i} \mathsf{Q}(k|x_{l,i}) h_{k,r}(x_{l,i})}{2} \Phi(-y_{l,i} f_{t-1}(x_{l,i}) - \eta)$$
$$+ \frac{1 - y_{l,i} \mathsf{Q}(k|x_{l,i}) h_{k,r}(x_{l,i})}{2} \Phi(-y_{l,i} f_{t-1}(x_{l,i}) + \eta).$$

In the case of exponential and logistic functions, the following upper bounds can then be derived.

**Lemma 1.** *For any $k \in [p]$, the following upper bound holds when $\Phi$ is the exponential or the logistic function:*

$$F(\alpha_{t-1} + \eta \mathbf{e}_{k,r}) \leq \max_{l \in [p]} \frac{Z_{t,l}}{m_l} \big[(1 - \epsilon_{t,l,k,r}) e^{-\eta} + \epsilon_{t,l,k,r} e^{\eta}\big],$$

*where $\epsilon_{t,l,k,r} = \frac{1}{2}\big[1 - \mathbb{E}_{i \sim \mathsf{D}_t(l,\cdot)}[y_{l,i} \mathsf{Q}(k|x_{l,i}) h_{k,r}(x_{l,i})]\big]$.*

For any $k$, function $\eta \mapsto (1 - \epsilon_{t,l,k,r}) e^{-\eta} + \epsilon_{t,l,k,r} e^{\eta}$ reaches its minimum for $\eta = \frac{1}{2} \log \frac{1 - \epsilon_{t,l,k,r}}{\epsilon_{t,l,k,r}}$. When the maximum is achieved for $l = k$, the solution coincides with the familiar expression of the step size $\eta_t = \frac{1}{2} \log \frac{1 - \epsilon_{t,k,r}}{\epsilon_{t,k,r}}$ used in ADABOOST.

**Q-function**. The conditional probability functions $\mathsf{Q}(k|\cdot)$ are crucial to the definition of our algorithm. As pointed out earlier, Q-ensembles can help achieve accurate solutions in some cases that cannot be realized using convex combinations. Furthermore, for any $k \in [p]$, since $\mathsf{D}_t(k,\cdot)$ is a distribution over the sample $S_k$, it is natural to assume that for any $j \neq k$ we have $\mathbb{E}_{s \sim \mathsf{D}_t(k,\cdot)}[\mathsf{Q}(k|x_{k,s})] \geq \mathbb{E}_{s \sim \mathsf{D}_t(k,\cdot)}[\mathsf{Q}(j|x_{k,s})]$. This implies the following lower bound: $\mathbb{E}_{s \sim \mathsf{D}_t(k,\cdot)}[\mathsf{Q}(k|x_{k,s})] \geq \frac{1}{p}$, which in turn implies $\overline{\gamma} \geq \frac{\gamma}{p}$, since for any $x \in \mathcal{X}$, $\sum_{j=1}^{p} \mathsf{Q}(j|x) = 1$. In the special case where all domains coincide, we have $\mathsf{Q}(k|x_{k,s}) = \frac{1}{p}$ for all $s$ and this lower bound is reached. At another extreme, when all domains admit distinct supports, we have $\mathsf{Q}(k|x_{k,s}) = 1$ for all $s \in [m_k]$ and thus $\overline{\gamma} = \gamma$.

---

MULTIBOOST$(S_1, \ldots, S_p)$
1  $\alpha_0 \leftarrow 0$
2  **for** $t \leftarrow 1$ **to** $T$ **do**
3      $f_{t-1} \leftarrow \sum_{l=1}^{p} \sum_{j=1}^{N_l} \alpha_{t-1,l,j} \mathsf{Q}(l|\cdot)\, h_{l,j}$
4      $\tilde{\Phi}_k \leftarrow \frac{1}{m_k} \sum_{i=1}^{m_k} \Phi(-y_{k,i} f_{t-1}(x_{k,i}))$
5      $\mathcal{K}_t \leftarrow \left\{ k \colon k \in \operatorname{argmax}_{k \in [p]} \tilde{\Phi}_k \right\}$
6      **for** $k \in \mathcal{K}_t$ **do**
7          $Z_{t,k} \leftarrow \sum_{i=1}^{m_k} \Phi'\big(-y_{k,i} f_{t-1}(x_{k,i})\big)$
8          **for** $i \leftarrow 1$ **to** $m_k$ **do**
9              $\mathsf{D}_t(k,i) \leftarrow \frac{\Phi'(-y_{k,i} f_{t-1}(x_{k,i}))}{Z_{t,k}}$
10     $(k,r) \leftarrow \operatorname*{argmax}_{k \in \mathcal{K}_t, r \in [N_k]} \frac{Z_{t,k}}{m_k}[1 - 2\epsilon_{t,k,r}]$
11     $\eta_t \leftarrow \operatorname{argmin}_{\eta > 0} F(\alpha_{t-1} + \eta \mathbf{e}_{k,r})$
12     $\alpha_t \leftarrow \alpha_{t-1} + \eta_t \mathbf{e}_{k,r}$
13 $f \leftarrow \sum_{l=1}^{p} \sum_{j=1}^{N_l} \alpha_{T,l,j} \mathsf{Q}(l|\cdot) h_{l,j}$
14 **return** $f$

---

**Figure 1:** Pseudocode of the MULTIBOOST algorithm. $\epsilon_{t,k,r} = \frac{1}{2}\big[1 - \mathbb{E}_{i \sim \mathsf{D}_t(k,\cdot)}[y_{k,i} \mathsf{Q}(k|x_{k,i}) h_{k,r}(x_{k,i})]\big]$ denotes the weighted error of $\mathsf{Q}(k|\cdot) h_{k,r}$.

In practice, we do not have access to the true conditional probabilities $\mathsf{Q}(k|\cdot)$. Instead, we can derive accurate estimates $\widehat{\mathsf{Q}}(k|\cdot)$ using large unlabeled samples from the source domains, the *label* used for training being simply the domain index. This can be done using algorithms such as conditional maximum entropy models (Berger et al., 1996) (or multinomial logistic regression), which benefit from strong theoretical guarantees (Mohri et al., 2018, Chapter 13), or other rich models based on neural networks.

**Other variants of MULTIBOOST**. In Appendix D, we present and discuss other variants of our algorithm, including their convergence guarantees.

# 4 Theoretical analysis

In this section, we present a theoretical analysis of our algorithm, including margin-based learning bounds for multiple-source Q-ensembles.

For any $k \in [p]$, define the family $\mathcal{G}_k$ as follows: $\mathcal{G}_k = \{Q(k|\cdot)\, h \colon h \in \mathcal{H}_k\}$. Then, the family of ensemble functions $\mathcal{F}$ that we consider can be defined as $\mathcal{F} = \mathrm{conv}(\bigcup_{k=1}^{p} \mathcal{G}_k)$. For any $\lambda \in \Delta$, let $\overline{\mathcal{D}}_\lambda$ be the distribution defined by $\mathcal{D}_\lambda = \sum_{k=1}^{p} \lambda_k \widehat{\mathcal{D}}_k$, where $\widehat{\mathcal{D}}_k$ is the empirical distribution associated to an i.i.d. sample $S_k$ drawn from $\mathcal{D}_k$. $\overline{\mathcal{D}}_\lambda$ is distinct from the distribution associated to $\mathcal{D}_\lambda$. We seek to derive margin-based bounds for ensemble functions $f \in \mathcal{F}$ with respect to target mixture distributions $\mathcal{D}_\lambda$, while the empirical data is from multiple source distributions $\mathcal{D}_k$, or from $\overline{\mathcal{D}}_\lambda$. Thus, these guarantees differ from standard learning bounds where the source and target distribution coincide.

For any distribution $\mathcal{D}$ over $\mathcal{X} \times \mathcal{Y}$ and $\rho > 0$, let $\mathcal{L}_\rho(\mathcal{D}, f)$ denote the expected $\rho$-margin loss of $f$ with respect to $\mathcal{D}$: $\mathcal{L}_\rho(\mathcal{D}, f) = \mathbb{E}_{(x,y)\sim\mathcal{D}_k}[\mathbb{I}(yf(x) \leq \rho)]$. We denote by $\mathfrak{R}_m(\mathcal{F})$ the Rademacher complexity of $\mathcal{F}$ for samples $S = (x_1, \ldots, x_m)$ of size $m$, defined by $\mathfrak{R}_m(\mathcal{F}) = \mathbb{E}_{S \sim \mathcal{D}^m}[\sup_{f \in \mathcal{F}} \sum_{i=1}^{m} \sigma_i f(x_i)]$, with $\sigma_i$s independent random variables taking values in $\{-1, +1\}$. The following gives a margin-bound for our multiple-source learning with Q-ensembles.

**Theorem 1.** *Fix $\rho > 0$. Then, for any $\delta > 0$, with probability at least $1 - \delta$ over the draw of a sample $S = (S_1, \ldots, S_p) \sim \mathcal{D}_1^{m_1} \otimes \cdots \otimes \mathcal{D}_p^{m_p}$, the following inequality holds for all ensemble functions $f = \sum_{t=1}^{T} \alpha_t Q(k_t|\cdot) h_t \in \mathcal{F}$ and all $\lambda \in \Delta$:*

$$\mathcal{L}(\mathcal{D}_\lambda, f) \leq \mathcal{L}_\rho(\overline{\mathcal{D}}_\lambda, f) + \sum_{k=1}^{p} \lambda_k \left[ \frac{2}{\rho} \max_{l \in [p]} \mathfrak{R}_{m_k}(\mathcal{H}_l) + \sqrt{\frac{\log \frac{p}{\delta}}{2m_k}} \right]. \tag{5}$$

Note that the complexity term depends only on the Rademacher complexities of the families of based predictors $\mathcal{H}_l$ and does not involve the domain classification function Q. Theorem 1 improves upon previous known bounds of Mohri et al. (2019). In particular, it provides a dimension-independent margin guarantee for the zero-one loss, while the bound of Mohri et al. (2019), when applied to the zero-one loss, is dimension-dependent (Mohri et al., 2019, Lemma 3). Additionally, our bound holds for Q-ensembles. Our margin bound can be straightforwardly generalized to hold uniformly for all $\rho \in (0, 1)$, at the cost of an additional mild term depending on $\log \log_2(2/\rho)$ (see Theorem 4 in Appendix E). For the ensemble functions returned by our algorithm, the bound applies to $x \mapsto \frac{f(x)}{\|\alpha\|_1}$. These learning guarantees suggest choosing $\alpha \geq 0$ and $\rho$ to minimize the following:

$$\max_{\lambda \in \Lambda} \sum_{k=1}^{p} \lambda_k \left\{ \mathbb{E}_{(x,y)\sim S_k} \left[ \mathbb{I}\left( y \sum_{t=1}^{T} \frac{\alpha_t}{\|\alpha\|_1} Q(k_t|\cdot) h_t(x) \leq \rho \right) \right] + \frac{r_k}{\rho} \right\},$$

where $r_k = 2 \max_{l \in [p]} \mathfrak{R}_{m_k}(\mathcal{H}_l)$. Choosing $\frac{1}{\rho} = \|\alpha\|_1$ and upper bounding the binary loss using a convex function $\Phi$, the problem can then be equivalently cast as the following unconstrained minimization, for example in the case of the exponential function:

$$\min_{\alpha \geq 0} \max_{\lambda \in \Lambda} \sum_{k=1}^{p} \lambda_k \left\{ \mathbb{E}_{(x,y)\sim S_k} \left[ \exp\left( -y \sum_{t=1}^{T} \alpha_t Q(k_t|\cdot) h_t(x) \right) \right] + r_k \|\alpha\|_1 \right\}.$$

Thus, the learning guarantees justify a posteriori an $\ell_1$-regularized version of our algorithm. It is straightforward to derive that extension of our algorithm and its pseudocode. This is similar to the extension to the $\ell_1$-ADABOOST (Rätsch et al., 2001) of the original unregularized ADABOOST (Freund and Schapire, 1997). Let us add that finer learning guarantees such as those for DEEP BOOSTING (Cortes et al., 2014) ones can be derived similarly here, which would lead to a weighted $\ell_1$-regularization, where the weights are the Rademacher complexities of the families $\mathcal{H}_k$. For the sake of simplicity, we do not detail that extension. Let us point out, however, that such an extension would also lead to an algorithm generalizing ADANET (Cortes et al., 2017) to multiple sources.

In some applications, prior knowledge can be used to constrain $\Lambda$ to a strict subset of the simplex $\Delta$. Finer margin-based learning guarantees can be derived to cover that scenario. Let $\overline{\Lambda}$ be the set of

vertices of a subsimplicial cover of $\Lambda$, that is a decomposition of a cover of $\Lambda$ into subsimplices and let $\overline{\Lambda}_\epsilon$ be the set of $\lambda$s $\epsilon$-close to $\overline{\Lambda}$ in $\ell_1$-distance. Then, the following guarantees hold.

**Theorem 2.** *Fix $\rho > 0$ and $\epsilon > 0$. Then, for any $\delta > 0$, with probability at least $1 - \delta$ over the draw of a sample $S = (S_1, \ldots, S_p) \sim \mathcal{D}_1^{m_1} \otimes \cdots \otimes \mathcal{D}_p^{m_p}$, each of the following inequalities holds:*
*1. for all ensemble functions $f = \sum_{t=1}^T \alpha_t \mathsf{Q}(k_t|\cdot)h_t \in \mathcal{F}$ and all $\lambda \in \overline{\Lambda}_\epsilon$:*

$$\mathcal{L}(\mathcal{D}_\lambda, h) \leq \mathcal{L}_\rho(\overline{\mathcal{D}}_\lambda, h) + \frac{2}{\rho} \max_{r \in [p]} \mathfrak{R}_\mathbf{m}(\mathcal{H}_r, \lambda) + \frac{3\epsilon}{2} + \sqrt{\sum_{k=1}^p \frac{\lambda_k^2}{2m_k} \log \frac{|\overline{\Lambda}|}{\delta}}. \qquad (6)$$

*2. for all ensemble functions $f = \sum_{t=1}^T \alpha_t \mathsf{Q}(k_t|\cdot)h_t \in \mathcal{F}$ and all $\lambda = \sum_{k=1}^p \mu_k \beta_k \in \mathrm{conv}(\overline{\Lambda})$:*

$$\mathcal{L}(\mathcal{D}_\lambda, h) \leq \mathcal{L}_\rho(\overline{\mathcal{D}}_\lambda, h) + \frac{2}{\rho} \sum_{l=1}^p \mu_k \max_{r \in [p]} \mathfrak{R}_\mathbf{m}(\mathcal{H}_r, \beta_l) + \sum_{k=1}^p \mu_k \sqrt{\sum_{l=1}^p \frac{\beta_{l,k}^2}{2m_k} \log \frac{|\overline{\Lambda}|}{\delta}}. \qquad (7)$$

The proof of the theorem and further discussion are presented in Appendix F. Here, $\mathfrak{R}_\mathbf{m}(\mathcal{H}_r, \lambda)$ is the $\lambda$-weighted Rademacher complexity of $\mathcal{H}_r$ (see Appendix F). For $\Lambda$ a strict subset of the simplex, these learning bounds suggest an algorithm with a regularization term of the form $\sum_{k=1}^p \lambda_k^2/m_k$. Our analysis and learning guarantees can be similarly extended to cover the multi-class setting.

## 5 Federated MULTIBOOST algorithm

In this section, we present an extension of our algorithm to the *federated learning* scenario, called FEDMULTIBOOST. The pseudocode is given in Figure 2. Federated learning is a distributed learning scenario where a global model is trained at the server level, based on data remaining at clients (Konečnỳ et al., 2016b,a; McMahan et al., 2017; Yang et al., 2019). Clients are typically mobile phones, network sensors, or other IoT devices. While the data remains at the clients, the trained model significantly benefits from user data, as demonstrated, for example, in next word prediction (Hard et al., 2018; Yang et al., 2018) and healthcare applications (Brisimi et al., 2018). We refer the reader to (Kairouz et al., 2021) for a more detailed survey of federated learning.

A widely used algorithm in this scenario is *federated averaging*, where the server trains the global model via stochastic updates from the clients (McMahan et al., 2017). It has been argued by Mohri et al. (2019), however, that minimizing the expected loss with respect to a particular training distribution, as done by standard federated learning algorithms, may be risk-prone and benefit only a specific subset of clients. This compromises fairness,

---

FEDMULTIBOOST$(S_1, S_2, \ldots, S_n)$
1  $\alpha_0 \leftarrow 0$
2  **for** $t \leftarrow 1$ **to** $T$ **do**
3      $f_{t-1} \leftarrow \sum_{l=1}^p \sum_{j=1}^{N_l} \alpha_{t-1,l,j} \mathsf{Q}(l|\cdot) h_{l,j}$
4      $C_{t,k} \leftarrow \text{SELECTCLIENT}(k, r)$
5      **for** $k \leftarrow 1$ **to** $p$ **do**
6          **for** $c \in C_{t,k}$ **do**
7              $\tilde{\Phi}_{k,c} \leftarrow \frac{1}{m_c} \sum_{i=1}^{m_c} \Phi(-y_{c,i} f_{t-1}(x_{c,i}))$
8              $Z_{t,c} \leftarrow \sum_{i=1}^{m_c} \Phi'(-y_{c,i} f_{t-1}(x_{c,i}))$
9      $\mathcal{K}_t \leftarrow \left\{ k \in [p] : k \in \mathrm{argmax}_{k \in [p]} \sum_{c \in C_{t,k}} \tilde{\Phi}_{k,c} \right\}$
10     $Z_{t,k} \leftarrow \sum_{c \in C_{t,k}} Z_{t,c}, \forall k \in [p]$
11     $\mathcal{H}_{t,k} \leftarrow \text{SELECTBASECLASSIFIER}(k, s)$
12     **for** $k \in \mathcal{K}_t$ **do**
13         **for** $c \in C_{t,k}$ **do**
14             **for** $i \leftarrow 1$ **to** $m_c$ **do**
15                 $\mathsf{D}_t(c, i) \leftarrow \frac{\Phi'(-y_{c,i} f_{t-1}(x_{c,i}))}{Z_{t,k}}$
16             $\beta_{t,c,r} \leftarrow [1 - 2\epsilon_{t,c,r}]$
17     $(k, r) \leftarrow \mathrm{argmax}_{k \in \mathcal{K}_t, r \in \mathcal{H}_{t,k}} \frac{Z_{t,k}}{\sum_{c \in C_{t,k}} m_c} \sum_{c \in C_{t,k}} \beta_{t,c,r}$
18     $\eta_t \leftarrow \eta_0/\sqrt{t}$
19     $\alpha_t \leftarrow \alpha_{t-1} + \eta_t \mathbf{e}_{k,r}$
20  $f \leftarrow \sum_{l=1}^p \sum_{j=1}^{N_l} \alpha_{T,l,j} \mathsf{Q}(l|\cdot) h_{l,j}$
21  **return** $f$

---

**Figure 2:** Pseudocode of the FEDMULTIBOOST algorithm. $\epsilon_{t,c,r}$ denotes the weighted error of $\mathsf{Q}(k|\cdot)h_{k,r}$, $\epsilon_{t,c,r} = \frac{1}{2}[1 - \mathbb{E}_{i \sim \mathsf{D}_t(c,\cdot)}[y_{c,i} \mathsf{Q}(k|x_{c,i})h_{k,r}(x_{c,i})]]$, where client $c$ belongs to domain $k$. cobalt steps are carried out by the server, red on the clients.

which is one of the critical questions in federated learning, where clients are often heterogeneous (Bickel et al., 1975; Hardt et al., 2016; Abay et al., 2020; Li et al., 2020). This issue can be further

accentuated when the set of participating clients during inference may significantly differ from those used in training. Such cases may occur, for example, when clients loose access to network connection during training, which results in distinct training and test distributions.

To overcome these problems, we adopt the *agnostic federated learning* approach suggested by Mohri et al. (2019) and Ro et al. (2021). Following the client-partition approach of Ro et al. (2021), let each client belong to one of $p$ domains. The domains can be, for example, based on the type of device or their geographical location. In this setting, the global model is optimized for all convex combinations of domain distributions as in the objective of MULTIBOOST (Equation 4), by taking the maximum over the mixtures weights $\lambda$. This ensures that the solution obtained is risk-averse and reduces the worst-case mismatch between the inference and training distributions. We propose FEDMULTIBOOST, a communication-efficient modification of MULTIBOOST, that can overcome the communication bottleneck in federated learning (Konečnỳ et al., 2016b).

Before presenting our algorithm, we first describe the learning scenario and the notation we adopt. Let $n$ be the number of clients, which in the cross-device setting can be in the order of millions. Let $m_c$ denote the number of samples in client $c$ and $(x_{c,i}, y_{c,i})$ denote the $i^{\text{th}}$ sample of client $c$. The server has access to a set of base-predictor classes $\mathcal{H}_k$ for $k \in [p]$ and a domain classifier $\mathsf{Q}$ that assigns for each sample $x$, the likelihood of it belonging to each of the $p$ domains. The base classifiers can be obtained by training on public data or training on client data with federated learning. The domain classifier $\mathsf{Q}$ can be trained using unlabeled client data.

Additional discussion of the algorithm as well as experimental results can be found in Appendix G. At each round, the algorithm randomly selects $r$ clients and $s$ base classifiers from each domain. Next, it chooses the best classifier out of the previously selected subset, based on the data from the subsampled clients. Since FEDMULTIBOOST operates only on a small set of base classifiers at each round, the algorithm is communication-efficient. At round $t$, the server needs to send $\mathsf{Q}$, $f_{t-1}$, and $s$ base classifiers to the selected clients. Hence, the server-to-client communication cost at round $t$ is $\mathcal{O}((t+s)K + K')$, where $K$ is the number of parameters in a single base classifier and $K'$ is the number of parameters required to specify $\mathsf{Q}$. Furthermore, the clients communicate only the aggregate statistics $\tilde{\Phi}_c, Z_{t,c}, \beta_{t,c,r}$ to the server and hence the client-to-server communication is $\mathcal{O}(p \cdot s)$ real numbers, which is independent of the number of parameters of the base classifiers. We note that FEDMULTIBOOST can be extended to the multi-class setting using techniques presented in Appendix L.

## 6    Experiments

In this section, we present experimental results for the MULTIBOOST algorithm on several multiple-source datasets. Our study is restricted to learning an ensemble of decision stumps $\mathcal{H}^{stumps}$ using the exponential surrogate loss $\Phi(u) = e^{-u}$. To estimate the probabilities $\mathsf{Q}(k|\cdot)$ for $k \in [p]$, we assigned the label $k$ to each sample from domain $k$ and used multinomial logistic regression. This step can be facilitated with unlabeled examples, as only their domain membership information is required.

We compare MULTIBOOST with a set of boosting-style algorithms that operate on the same hypotheses class $\mathcal{H}^{stumps}$. Those include ADABOOST-$k$ for $k \in [p]$ and ADABOOST-all. The former is a standard ADABOOST algorithm trained only on a single source $S_k$ and the latter is ADABOOST trained on the union of all sources $\cup_{k=1}^p S_k$. It is natural to compare MULTIBOOST against ADABOOST-all, since both of them have access to all sources during training. The difference is that while ADABOOST-all treats $\cup_{k=1}^p S_k$ as a single dataset, MULTIBOOST trains base learners separately for each source and weights examples by domain probabilities $\mathsf{Q}(k|\cdot)$. We used $T = 100$ boosting steps for all benchmarks. $T$ up to 1,000 were explored, but did not change any results significantly.

We also compare our results with the a multiple-source adaptation algorithm, DMSA, designed for a scenario where no labeled data is available, and where only an accurate predictor per domain and unlabeled data are supplied. DMSA was shown to outperform other algorithms designed for that scenario. To apply DMSA, we use the domain predictors ADABOOST-$k$, $k \in [p]$.

We report classification errors on the test data for various mixtures of the source distributions $\lambda_1, \ldots, \lambda_p$, including: errors for $\lambda$ on the simplex $\Delta$ edges (errors on each domain separately); errors on the uniform mixture $\forall k : \lambda_k = \frac{1}{p}$; agnostic error, which is the maximum error across all sources. The errors and their standard deviations are reported based on 10-fold cross validation. Each source

$S_k$ is independently split into 10 folds $S_k^1, \ldots, S_k^{10}$. For the $i$-th cross-validation step, the test set is $\{S_1^i, \ldots, S_p^i\}$, while the rest is used for training.

Datasets and preprocessing steps used are described below with additional dataset details provided in Appendix H. Note, that all datasets are public and do not contain any personal identifiers or offensive information. The experiments were performed on Linux and Mac workstations with Quad-Core Intel Core i7 2.9 GHz and Intel Xeon 2.20 GHz respectively. The time taken to perform 10-fold cross-validation varies per dataset: from roughly 1 hour on CPU for all benchmarks on tabular data to 8 hours on CPU for all benchmarks on digits data. The run-times of a single MULTIBOOST iteration is almost identical to that of ADABOOST-all when a closed form solution is used for step size $\eta_t$ (line 12 in Figure 1) for the exponential surrogate loss function. Alternatively, for some experiments we used line search with 100 steps. If training time is critical, one can use $\eta_t = C/\sqrt{t}$ instead, at the cost of slightly worse convergence guarantee. Convergence plots are illustrated in Appendix I.

**Sentiment analysis.** The *Sentiment* dataset (Blitzer et al., 2007) consists of text reviews and rating labels for products sold on Amazon in various categories. We selected $p = 3$ sources from this dataset, where $k = 1$ corresponds to `books`, $k = 2$ means `dvd`, and $k = 3$ is `electronics`.

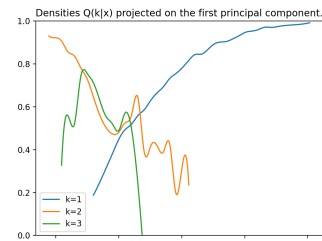

We converted the 5-star rating to binary labels, assigning $y = +1$ for a positive review and $y = -1$ for a negative one. Neutral reviews with 3-star labels are removed from the dataset. The benchmarks are trained using bigram features generated from the raw text reviews. Since the *Sentiment* dataset contains only 2,000 instances for each domain, we used random train/test splits with 10 different seeds instead of the 10-fold cross-validation. To illustrate the importance of the conditional domain probabilities $Q(K|x)$, for each domain, we illustrate its values along the projections of $x$ onto the first principal component of the joint *Sentiment* dataset in Figure 3. The figure shows that our domain classifiers are able to provide coherent separation between the three domains and narrow the applicability of the base classifiers.

**Figure 3:** Mean values of $Q(k|\cdot_k)$ for the three domains of the Sentiment data projected on the first principal component of the joint data.

In Appendix K, we further discuss the importance of obtaining a high-quality Q function and illustrate the performance degradation resulting from a lower-quality Q function.

**Digits recognition.** For this problem, we aggregated 32x32 pixel handwritten digits images from $p = 3$ sources: *MNIST* ($k = 1$), *SVHN* ($k = 2$), *MNIST-M* ($k = 3$). We compared algorithms on two binary classification problems: digits 4 vs. 9 and digits 1 vs. 7.

**Object recognition.** We divided images of clothes items from *Fashion-MNIST* (Xiao et al., 2017) dataset into two classes: tops and bottoms from $p = 3$ sources. $k = 1$ consists of `t-shirts, pullovers, trousers`; $k = 2$ consists of `dresses, coats, sandals`; $k = 3$ consists of `shirts, bags, sneakers, ankle-boots`.

**Tabular data.** In addition to text and images, we tested MULTIBOOST on tabular data. We used the *Adult* dataset (Kohavi, 1996), which consists of numerical and categorical features describing an individual's socioeconomic characteristics, given the task to predict if an person's income exceeds USD $50,000$. We divided this dataset into $p = 3$ sources based on individual's educational background. Source $k = 1$ consisted of individuals with a university degree (BSc, MS or PhD), source $k = 1$ those with only a High School diploma, and source $k = 2$, none of the above.

**Discussion** As can be seen from Table 4, MULTIBOOST provides agnostic and uniform errors that are significantly better than the baselines on all datasets. While ADABOOST-k predictors on digits recognition and object recognition tasks each show the lowest error on their specific domains $\mathcal{D}_1, \mathcal{D}_2, \mathcal{D}_3$, they fail to generalize to other target distribution in $\text{conv}\{\mathcal{D}_1, \mathcal{D}_2, \mathcal{D}_3\}$. Moreover, as suggested by the discussion in Section 2, the standard convex combination of domain-specific predictors (ADABOOST-all) also performs poorly on several target distributions in $\text{conv}\{\mathcal{D}_1, \mathcal{D}_2, \mathcal{D}_3\}$, such as the uniform and agnostic targets. As the experimental results confirm, MULTIBOOST, by leveraging domain predictors $Q(k|\cdot)$ and the agnostic loss, is able to produce an ensemble of domain-specific predictors that generalizes to different targets. Appendix I provides additional illustrations of $Q(k|\cdot)$ by projecting each domain on the first principal component of the joint data.

Moreover, for tabular data and the digits (4 vs. 9) classification problem, MULTIBOOST benefits from positive knowledge transfer and obtains an error on individual domains that is even smaller than that

**Table 1:** Test errors for multiple benchmarks.

| Algorithm | Error-1 | Error-2 | Error-3 | Error-Uniform | Error-Agnostic |
|---|---|---|---|---|---|
| Sentiment Analysis | | | | | |
| ADABOOST-1 | **0.326 ± 0.019** | 0.300 ± 0.008 | 0.360 ± 0.017 | 0.329 ± 0.017 | 0.360 ± 0.019 |
| ADABOOST-2 | 0.354 ± 0.020 | **0.266 ± 0.009** | 0.336 ± 0.023 | 0.318 ± 0.019 | 0.357 ± 0.019 |
| ADABOOST-3 | 0.402 ± 0.015 | 0.334 ± 0.008 | **0.258 ± 0.015** | 0.331 ± 0.016 | 0.402 ± 0.018 |
| ADABOOST-all | 0.354 ± 0.020 | 0.325 ± 0.011 | 0.313 ± 0.022 | 0.324 ± 0.021 | 0.354 ± 0.016 |
| DMSA | 0.332 ± 0.021 | 0.308 ± 0.017 | 0.314 ± 0.015 | 0.318 ± 0.019 | 0.332 ± 0.021 |
| MULTIBOOST | 0.332 ± 0.027 | 0.288 ± 0.018 | 0.284 ± 0.027 | **0.301 ± 0.027** | **0.332 ± 0.024** |
| Digits Recognition (4 vs. 9) | | | | | |
| ADABOOST-1 | **0.044 ± 0.007** | 0.615 ± 0.012 | 0.476 ± 0.022 | 0.379 ± 0.008 | 0.615 ± 0.012 |
| ADABOOST-2 | 0.455 ± 0.014 | 0.299 ± 0.011 | 0.504 ± 0.015 | 0.420 ± 0.011 | 0.504 ± 0.015 |
| ADABOOST-3 | 0.549 ± 0.034 | 0.488 ± 0.015 | 0.300 ± 0.013 | 0.446 ± 0.013 | 0.549 ± 0.034 |
| ADABOOST-all | 0.060 ± 0.009 | 0.374 ± 0.015 | 0.353 ± 0.012 | 0.262 ± 0.009 | 0.374 ± 0.015 |
| DMSA | 0.069 ± 0.005 | 0.351 ± 0.012 | 0.310 ± 0.011 | 0.243 ± 0.009 | 0.351 ± 0.015 |
| MULTIBOOST | 0.096 ± 0.008 | **0.283 ± 0.028** | **0.246 ± 0.014** | **0.209 ± 0.013** | **0.284 ± 0.027** |
| Digits Recognition (1 vs. 7) | | | | | |
| ADABOOST-1 | **0.005 ± 0.002** | 0.613 ± 0.007 | 0.519 ± 0.012 | 0.379 ± 0.004 | 0.613 ± 0.007 |
| ADABOOST-2 | 0.431 ± 0.022 | **0.252 ± 0.009** | 0.479 ± 0.012 | 0.387 ± 0.010 | 0.479 ± 0.012 |
| ADABOOST-3 | 0.680 ± 0.031 | 0.490 ± 0.014 | **0.244 ± 0.012** | 0.474 ± 0.013 | 0.680 ± 0.031 |
| ADABOOST-all | 0.014 ± 0.003 | 0.286 ± 0.010 | 0.306 ± 0.012 | 0.202 ± 0.005 | 0.306 ± 0.011 |
| DMSA | 0.012 ± 0.003 | 0.288 ± 0.017 | 0.286 ± 0.015 | 0.195 ± 0.013 | 0.288 ± 0.017 |
| MULTIBOOST | 0.026 ± 0.004 | 0.261 ± 0.013 | 0.257 ± 0.015 | **0.181 ± 0.005** | **0.261 ± 0.011** |
| Objects Recognition (Fashion-MNIST) | | | | | |
| ADABOOST-1 | **0.015 ± 0.003** | 0.251 ± 0.026 | 0.602 ± 0.028 | 0.288 ± 0.017 | 0.602 ± 0.028 |
| ADABOOST-2 | 0.435 ± 0.007 | **0.015 ± 0.002** | 0.169 ± 0.012 | 0.173 ± 0.003 | 0.435 ± 0.007 |
| ADABOOST-3 | 0.311 ± 0.018 | 0.097 ± 0.005 | **0.014 ± 0.002** | 0.140 ± 0.006 | 0.311 ± 0.018 |
| ADABOOST-all | 0.036 ± 0.004 | 0.020 ± 0.002 | 0.025 ± 0.003 | 0.027 ± 0.002 | 0.036 ± 0.004 |
| DMSA | 0.033 ± 0.008 | 0.015 ± 0.002 | 0.022 ± 0.003 | 0.023 ± 0.007 | 0.033 ± 0.009 |
| MULTIBOOST | 0.028 ± 0.003 | 0.015 ± 0.003 | 0.022 ± 0.002 | **0.021 ± 0.001** | **0.028 ± 0.003** |
| Tabular Data (Adult Data) | | | | | |
| ADABOOST-1 | 0.201 ± 0.012 | 0.249 ± 0.021 | 0.215 ± 0.014 | 0.222 ± 0.012 | 0.249 ± 0.021 |
| ADABOOST-2 | 0.298 ± 0.011 | **0.131 ± 0.009** | 0.142 ± 0.006 | 0.190 ± 0.007 | 0.298 ± 0.011 |
| ADABOOST-3 | 0.225 ± 0.015 | 0.138 ± 0.010 | 0.133 ± 0.011 | 0.165 ± 0.008 | 0.225 ± 0.015 |
| ADABOOST-all | 0.221 ± 0.013 | 0.134 ± 0.007 | 0.131 ± 0.010 | 0.155 ± 0.004 | 0.221 ± 0.013 |
| DMSA | 0.195 ± 0.014 | 0.137 ± 0.005 | 0.131 ± 0.008 | 0.154 ± 0.005 | 0.195 ± 0.014 |
| MULTIBOOST | **0.190 ± 0.014** | 0.132 ± 0.008 | **0.130 ± 0.009** | **0.150 ± 0.005** | **0.190 ± 0.014** |

of the domain-specific predictors. From Figure 6 in Appendix I it can be seen that the $\alpha$-weight for the MULTIBOOST classifier in the 4 vs. 9 problem is heavily centered on $\mathcal{D}_2$, but contributions from $\mathcal{D}_1$ and $\mathcal{D}_3$ adds to the stronger performance. The same figure also illustrates the $\alpha$-mass distribution for the other classification tasks. The standard convex ensemble ADABOOST-all, however, does not seem to exhibit the positive transfer property. Similar conclusions carry over for other target distributions in $\mathrm{conv}\{\mathcal{D}_1, \mathcal{D}_2, \mathcal{D}_3\}$, for the sake of simplicity we do not present them here.

## 7 Conclusion

We presented a new boosting algorithm, MULTIBOOST, for a multiple-source scenario – a common setting in domain adaptation problems – where the target distribution can be any mixture of the source distributions. We showed that our algorithm benefits from strong theoretical guarantees and exhibits favorable empirical performance. We also highlighted the extension of our work to the federated learning scenario, which is a critical distributed learning setting in modern applications.

## Acknowledgments

This work was partly supported by NSF CCF-1535987, NSF IIS-1618662, and a Google Research Award.

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
