# Contents of Appendix

# A    Previous work: more detailed discussion

Domain adaptation with a single source domain and a target domain has been widely studied (Ben-David et al., 2007; Blitzer et al., 2008; Mansour et al., 2009b; Ben-David et al., 2010; Cortes and Mohri, 2014; Cortes et al., 2015; Wang et al., 2019b) and has applications to several fields ranging from acoustic modelling (Liao, 2013) to object recognition (Torralba and Efros, 2011). It has been studied in unsupervised settings with unlabeled target domain examples (Gong et al., 2012; Long et al., 2015; Ganin and Lempitsky, 2015), in supervised settings with the aid of labeled target domain examples (Yang et al., 2007; Hoffman et al., 2013; Girshick et al., 2014; Motiian et al., 2017b), and in semi-supervised settings where both labeled and unlabeled target examples are available (Tzeng et al., 2015; Saito et al., 2019).

In a wide variety of applications, the learner has access to information from multiple sources. Such problems are often referred to as *multiple-source adaptation*. Multiple-source adaptation problems, where the learner has access to unlabeled source data together with predictors that are trained for each particular domain has been formalized in Mansour et al. (2008, 2009a); Hoffman et al. (2018). There are other multiple-source adaptation scenarios, where labeled examples are available from multiple sources and unlabeled or labeled examples are available from the target domain. This includes adversarial training, which has been studied by Motiian et al. (2017a); Pei et al. (2018); Zhao et al. (2018); Xu et al. (2018). Algorithms for learning from multiple untrusted sources have been proposed by Konstantinov and Lampert (2019). Another related problem is *domain generalization* (Pan and Yang, 2009; Muandet et al., 2013; Xu et al., 2014), where information from multiple sources is used to obtain a predictor that generalizes to a previously unseen domain.

There are various algorithms, successfully applying boosting with multiple sources to domain adaptation and transfer learning problems, that have inspired our analysis. The TRADABOOST (Dai et al., 2007) algorithm, having a set of weak learners trained on the source domain, at every boosting round selects those that minimize the error on the target domain. In case of multiple sources and a single target, Yao and Doretto (2010) developed MULTISOURCETRADABOOST algorithm that trains weak learners on the union of each of the sources and the target, thus reducing the negative knowledge transfer effect. These algorithms have been further improved and widely adopted in practice (Yuan et al., 2017; Cheng et al., 2013; Zhang et al., 2014). Another approach that uses multi-view ADABOOST for single and multi-source domain adaptation was proposed by Xu and Sun (2012, 2011). They divide the feature space into two *views* based on the source and target; at each boosting step, two weak learners are trained on these views and the sample distribution is updated according to the errors on the target domain.

A number of experimental studies have shown the benefits of having an ensemble of weak learners for multi-task learning and domain adaptation problems. Moreover, in certain cases the boosting approach can outperform traditional methods. For example, Huang et al. (2010, 2012) showed that by selecting a weak learner jointly with a feature that is predictive across multiple domains at every boosting step, one can achieve higher accuracy than standard transfer learning methods. Moreover, the margin provided by boosting-style algorithms can aid in transfer learning where target domain is unlabelled. Habrard et al. (2013) have developed an algorithm that jointly minimizes the the source domain error and margin violation proportion on the target domain.

Wang et al. (2019a) have demonstrated that boosting classifiers from different domains can be done online and showed efficient algorithm for the ADABOOST-style sample distribution updates. For certain types of high-dimensional data, such as images and text, boosting may be not as efficient as other multi-task learning methods. However, a number of works such as Taherkhani et al. (2020) and Becker et al. (2013) have shown that multi-source boosting can be combined with Deep Neural Networks for multi-task learning on large scale datasets.

In the context of neural networks, the idea of using domain probabilities when combining experts, also termed *gating networks*, goes back to Hampshire and Waibel (1992) and Jacobs et al. (1991).

Agnostic loss has been used in several machine learning problems. Namkoong and Duchi (2016); Levy et al. (2020) proposed efficient algorithms to minimize agnostic loss in the context of distributionally robust optimization. Agnostic loss in federated learning has been studied by Mohri et al. (2019); Hamer et al. (2020); Ro et al. (2021), who provided theoretical guarantees and algorithms. Lahoti et al. (2020) used agnostic loss to achieve fairness in machine learning models.

# B  Proof of the Propositions 1 and 2

This section contains the proofs for Proposition 1 and Proposition 2 discussed in Section 2.

**Proposition 1.** *There exist distributions $\mathcal{D}_1$ and $\mathcal{D}_2$ and hypotheses $h_1$ and $h_2$ with $\mathcal{L}(\mathcal{D}_1, h_1) = 0$ and $\mathcal{L}(\mathcal{D}_2, h_2) = 0$ such that $\mathcal{L}\big(\frac{1}{2}(\mathcal{D}_1 + \mathcal{D}_2), \alpha h_1 + (1 - \alpha)h_2\big) \geq \frac{1}{2}$ for any $\alpha \in [0, 1]$,*

*Proof.* Consider the case where $\mathcal{X}$ is reduced to two elements, $\mathcal{X} = \{a_1, a_2\}$, where $\mathcal{D}_1$ is the point mass on $a_1$, $\mathcal{D}_2$ the point mass on $a_2$, and where the target labeling function is $f$ defined by $f(a_1) = +1$, $f(a_2) = -1$. Let $h_1$ be defined by $h_1(a_1) = h_1(a_2) = +1$ and and $h_2$ by $h_2(a_1) = h_2(a_2) = -1$.

Then, $h_1$ is a perfect predictor for the first domain since $\mathcal{L}(\mathcal{D}_1, h_1) = \mathbb{I}(h_1(a_1)f(a_1) \leq 0) = 0$, and $h_2$ is a perfect predictor for the second domain since $\mathcal{L}(\mathcal{D}_2, h_2) = \mathbb{I}(h_2(a_2)f(a_2) \leq 0) = 0$. However, for any $\alpha \in [0, 1]$, we have

$$\mathcal{L}\left(\frac{1}{2}(\mathcal{D}_1 + \mathcal{D}_2), \alpha h_1 + (1 - \alpha)h_2\right) = \frac{1}{2}\,\mathbb{I}(2\alpha - 1 \leq 0) + \frac{1}{2}\,\mathbb{I}(1 - 2\alpha \leq 0) \geq \frac{1}{2}.$$

This concludes the proof of Proposition 1. $\qquad\square$

**Proposition 2.** *For the same distributions $\mathcal{D}_1$ and $\mathcal{D}_2$ and hypotheses $h_1$ and $h_2$ as in Proposition 1, the equality $\mathcal{L}(\mathcal{D}_\lambda, (\alpha \mathsf{Q}(1|\cdot)h_1 + (1 - \alpha)\mathsf{Q}(2|\cdot)h_2)) = 0$ holds for any $\lambda \in \Delta$ and any $\alpha \in (0, 1)$.*

*Proof.* For the counterexample of Proposition 1, for any $\alpha \in (0, 1)$, the Q-ensemble $f(x) = (\alpha \mathsf{Q}(1|x)h_1(x) + (1 - \alpha)\mathsf{Q}(2|x)h_2(x))$ admits no loss with respect to any target distribution $\mathcal{D}_\lambda$:

$$\mathcal{L}(\mathcal{D}_\lambda, f) = (\lambda\,\mathbb{I}(f(a_1) \leq 0) + (1 - \lambda)\,\mathbb{I}(-f(a_2) \leq 0))$$
$$= \lambda(\mathbb{I}(\alpha \leq 0) + (1 - \lambda)\,\mathbb{I}((1 - \alpha) \leq 0)) = 0,$$

since $\mathsf{Q}(1|a_1) = \mathsf{Q}(2|a_2) = 1$ and $\mathsf{Q}(2|a_1) = \mathsf{Q}(1|a_2) = 0$.

This concludes the proof of Proposition 2. $\qquad\square$

# C  Proof of Lemma 1

**Lemma 1.** *For any $k \in [p]$, the following upper bound holds when $\Phi$ is the exponential or the logistic function:*

$$F(\alpha_{t-1} + \eta\mathbf{e}_{k,r}) \leq \max_{l \in [p]} \frac{Z_{t,l}}{m_l}\Big[(1 - \epsilon_{t,l,k,r})e^{-\eta} + \epsilon_{t,l,k,r}e^{\eta}\Big],$$

*where $\epsilon_{t,l,k,r} = \frac{1}{2}\big[1 - \mathbb{E}_{i \sim \mathsf{D}_t(l,\cdot)}[y_{l,i}\mathsf{Q}(k|x_{l,i})h_{k,r}(x_{l,i})]\big]$.*

*Proof.* In the special case where $\Phi = \exp$, we have:

$$\Phi(-y_{l,i}f_{t-1}(x_{l,i}) - \eta\,y_{l,i}\mathsf{Q}(k|x_{l,i})h_{k,r}(x_{l,i}))$$
$$\leq \frac{1 + y_{l,i}\mathsf{Q}(k|x_{l,i})h_{k,r}(x_{l,i})}{2}e^{-y_{l,i}f_{t-1}(x_{l,i})}e^{-\eta} + \frac{1 - y_{l,i}\mathsf{Q}(k|x_{l,i})h_{k,r}(x_{l,i})}{2}e^{-y_{l,i}f_{t-1}(x_{l,i})}e^{\eta}$$
$$= \frac{1 + y_{l,i}\mathsf{Q}(k|x_{l,i})h_{k,r}(x_{l,i})}{2}\mathsf{D}_t(l,i)Z_{t,l}e^{-\eta} + \frac{1 - y_{l,i}\mathsf{Q}(k|x_{l,i})h_{k,r}(x_{l,i})}{2}\mathsf{D}_t(l,i)Z_{t,l}e^{\eta}.$$

Thus, we have:

$$F(\alpha_{t-1} + \eta\mathbf{e}_{k,r}) = \max_{l \in [p]} \frac{1}{m_l}\sum_{i=1}^{m_l}\Phi(-y_{l,i}f_{t-1}(x_{l,i}) - \eta\,y_{l,i}\mathsf{Q}(k|x_{l,i})h_{k,r}(x_{l,i}))$$
$$\leq \max_{l \in [p]} \frac{Z_{t,l}}{m_l}\sum_{i=1}^{m_l}\frac{1}{m_l}\sum_{i=1}^{m_l}\frac{1 + y_{l,i}\mathsf{Q}(k|x_{l,i})h_{k,r}(x_{l,i})}{2}\mathsf{D}_t(l,i)e^{-\eta}$$
$$+ \frac{1 - y_{l,i}\mathsf{Q}(k|x_{l,i})h_{k,r}(x_{l,i})}{2}\mathsf{D}_t(l,i)e^{\eta}$$
$$= \max_{l \in [p]} \frac{Z_{t,l}}{m_l}\Big[(1 - \epsilon_{t,l,k,r})e^{-\eta} + \epsilon_{t,l,k,r}e^{\eta}\Big].$$

The proof is similar in the case of the logistic function. $\qquad\square$

# D Other variants of MULTIBOOST

As already mentioned, instead of the maximum, the *softmax* function $(x_1, \ldots, x_k) \mapsto \frac{1}{\mu} \log(\sum_{k=1}^{p} e^{\mu x_k})$ can be used in the definition of the algorithm, modulo an approximation that can be controlled via the parameter $\mu > 0$. Using the *softmax* not only leads to a differentiable objective, but also makes the algorithm focus on several top most difficult domains instead of the single most difficult one, thereby offering a useful trade-off in some applications.

Another variant of the algorithm with also a differentiable objective function consists of simply upper-bounding the maximum by a sum:

$$F_{\text{sum}}(\alpha) = \sum_{k=1}^{p} \frac{1}{m_k} \sum_{i=1}^{m_k} \Phi\left(-y_{k,i} \sum_{l=1}^{p} \sum_{j=1}^{N_l} \alpha_{l,j} \mathsf{Q}(l|x_{k,i}) h_{l,j}(x_{k,i})\right). \tag{8}$$

It is straightforward to show that, as with the maximum-based objective, our weak learning assumption implies that, at each round, there exists a coordinate direction along which each active function $F_k$ decreases. Furthermore, our comments and analysis in the maximum case regarding the Q-function and lower bounds on the edge similarly hold here.

In the following, we present convergence guarantees for the $F_{\text{sum}}$ objective. A similar guarantee with the same proof holds for the softmax variant of MULTIBOOST.

**Theorem 3.** *Assume that $\Phi$ is twice differentiable and $\Phi''(u) \geq 0$ for all $u \in \mathbb{R}$. Let $F = F_{\text{sum}}$, then, projected coordinate descent applied to $F(\alpha)$ converges to the optimal solution $\alpha^*$ of $\min_{\alpha \geq 0} F(\alpha)$. If further $\Phi$ is strongly convex on the path of the iterates $\alpha_t$, then there exist $\tau > 0$ and $\gamma > 0$ such that for all $t > \tau$:*

$$F(\alpha_{t+1}) - F(\alpha^*) \leq \left(1 - \frac{1}{\gamma}\right)(F(\alpha_t) - F(\alpha^*)). \tag{9}$$

*Proof.* We show that $F_{\text{sum}}$ can be represented as $F_{\text{sum}}(\alpha) = G(\mathbf{H}\alpha)$, such that $\nabla^2 G(\mathbf{H}\alpha)$ is positive definite for all $\alpha$ and apply Theorem 2.1 in Luo and Tseng (1992) to obtain the convergence guarantees. Let $\mathbf{H}$ be the matrix whose row indexes are $\{(k,i)\colon k \in [p], i \in [m_k]\}$ and whose column indexes are $\{(l,j)\colon l \in [p], j \in [N_k]\}$. Define matrix $\mathbf{H}$ by $\mathbf{H}_{(k,i),(l,j)} = -y_{k,i}\mathsf{Q}(l|x_{k,i})h_{l,j}(x_{k,i})$. Let $\mathbf{e}_{(k,i)}$ be the $(k,i)$-th unit vector, then for any $\alpha$:

$$\mathbf{e}_{(k,i)}^{\top} \mathbf{H}\alpha = -y_{k,i} \sum_{l=1}^{p} \sum_{j=1}^{N_l} \alpha_{l,j} \mathsf{Q}(l|x_{k,i}) h_{l,j}(x_{k,i}). \tag{10}$$

Define the function $G$ as follows for all $\mathbf{u}$:

$$G(\mathbf{u}) = \sum_{k=1}^{p} \frac{1}{m_k} \sum_{i=1}^{m_k} \Phi\left(-\mathbf{e}_{(k,i)}^{\top} \mathbf{u}\right). \tag{11}$$

By definition, we have $F_{\text{sum}}(\alpha) = G(\mathbf{H}\alpha)$. Moreover, $G$ is twice differentiable and $\nabla^2 G(\mathbf{u})$ is a diagonal matrix with diagonal entries $\frac{1}{m_k}\Phi''(-\mathbf{e}_{(k,i)}^{\top}\mathbf{u}) \geq 0$. Thus, $\nabla^2 G(\mathbf{u})$ is positive definite for all $\alpha \geq 0$. Thus, Theorem 2.1 in Luo and Tseng (1992) holds for the optimization problem

$$\min_{\alpha \geq 0} G(\mathbf{H}\alpha), \tag{12}$$

which guarantees the convergence rate of the coordinate descent for $F_{\text{sum}}$. If further $F$ is is strongly convex over the sequence of $\alpha_t$s, then, by Luo and Tseng (1992)[page 26], the inequality:

$$F(\alpha_{t+1}) - F(\alpha^*) \leq \left(1 - \frac{1}{\gamma}\right)(F(\alpha_t) - F(\alpha^*))$$

holds for the projected coordinate method based on the best direction at each round, as with the Gauss-Southwell method. $\square$

Note, that the proof can be extended straightforwardly to a regularized $F_{\text{sum}}$ objective, simply by considering $F_{\text{sum}}(\alpha) = G(\mathbf{H}\alpha) + \beta^{\top}\alpha$ in the proof for some $\beta \geq 0$.

# E  Proofs of Theorem 1 and Theorem 4

**Theorem 1.** *Fix $\rho > 0$. Then, for any $\delta > 0$, with probability at least $1 - \delta$ over the draw of a sample $S = (S_1, \ldots, S_p) \sim \mathcal{D}_1^{m_1} \otimes \cdots \otimes \mathcal{D}_p^{m_p}$, the following inequality holds for all ensemble functions $f = \sum_{t=1}^{T} \alpha_t \mathsf{Q}(k_t|\cdot) h_t \in \mathcal{F}$ and all $\lambda \in \Delta$:*

$$\mathcal{L}(\mathcal{D}_\lambda, f) \leq \mathcal{L}_\rho(\overline{\mathcal{D}}_\lambda, f) + \sum_{k=1}^{p} \lambda_k \left[ \frac{2}{\rho} \max_{l \in [p]} \mathfrak{R}_{m_k}(\mathcal{H}_l) + \sqrt{\frac{\log \frac{p}{\delta}}{2 m_k}} \right]. \tag{5}$$

*Proof.* Fix $\lambda \in \Delta$ and $\delta > 0$. For any $k \in [p]$, by the standard Rademacher complexity margin bound for $\mathcal{F}$ (Mohri et al., 2018)[Theorem 5.8], with probability at least $1 - \delta$, the following inequality holds for all $f \in \mathcal{F}$:

$$\mathcal{L}(\mathcal{D}_k, f) \leq \mathcal{L}_\rho(\widehat{\mathcal{D}}_k, f) + \frac{2}{\rho} \mathfrak{R}_{m_k}(\mathcal{F}) + \sqrt{\frac{\log \frac{1}{\delta}}{2 m_k}}.$$

By the union bound, the following inequalities hold simultaneously for all $k \in [p]$:

$$\mathcal{L}(\mathcal{D}_k, f) \leq \mathcal{L}_\rho(\widehat{\mathcal{D}}_k, f) + \frac{2}{\rho} \mathfrak{R}_{m_k}(\mathcal{F}) + \sqrt{\frac{\log \frac{p}{\delta}}{2 m_k}}.$$

Multiplying each by $\lambda_k$ and summing them up yields:

$$\mathcal{L}(\mathcal{D}_\lambda, f) \leq \mathcal{L}_\rho(\overline{\mathcal{D}}_\lambda, f) + \sum_{k=1}^{p} \lambda_k \left[ \frac{2}{\rho} \mathfrak{R}_{m_k}(\mathcal{F}) + \sqrt{\frac{\log \frac{p}{\delta}}{2 m_k}} \right].$$

Since the Rademacher of a family coincides with that of its convex hull (Mohri et al., 2018), we have $\mathfrak{R}_{m_k}(\mathcal{F}) = \mathfrak{R}_{m_k}\left( \bigcup_{k=1}^{p} \mathcal{G}_k \right) \leq \max_{l \in [p]} \mathfrak{R}_{m_k}(\mathcal{G}_l)$. We will show that the following inequality holds for any $k \in p$: $\mathfrak{R}_{m_k}(\mathcal{G}_l) \leq 2 \mathfrak{R}_{m_k}(\mathcal{H}_l)$. Note that we can write for any $h \in \mathcal{H}_k$: $\mathsf{Q}(l|\cdot) h = \frac{1}{4} \left[ (\mathsf{Q}(l|\cdot) + h)^2 - (\mathsf{Q}(l|\cdot) - h)^2 \right]$. Thus, since the Rademacher complexity of a sum can be bounded by the sum of the Rademacher complexities, we have:

$$\mathfrak{R}_{m_k}(\mathcal{G}_l) \leq \frac{1}{4} \mathfrak{R}_{m_k}\left( \left\{ (\mathsf{Q}(l|\cdot) + h)^2 \colon h \in \mathcal{H}_l \right\} \right) + \frac{1}{4} \mathfrak{R}_{m_k}\left( \left\{ (\mathsf{Q}(l|\cdot) - h)^2 \colon h \in \mathcal{H}_l \right\} \right).$$

Now, functions $\mathsf{Q}(l|\cdot) + h$ and $\mathsf{Q}(l|\cdot) - h$ both take values in $[-1, 2]$ and the function $x \mapsto \frac{1}{4} x^2$ is 1-Lipschitz on $[-1, 2]$ since the absolute value of its derivative $|x|/2$ reaches it maximum at $x = 2$. Thus, by Talagrand's contraction lemma (Ledoux and Talagrand, 1991), we have

$$\mathfrak{R}_{m_k}(\mathcal{G}_l) \leq \mathfrak{R}_{m_k}(\{(\mathsf{Q}(l|\cdot) + h) \colon h \in \mathcal{H}_l\}) + \mathfrak{R}_{m_k}(\{(\mathsf{Q}(l|\cdot) - h) \colon h \in \mathcal{H}_l\}).$$

Now, these Rademacher complexities can be straightforwardly analyzed as follows:

$$
\begin{aligned}
\mathfrak{R}_{m_k}(\{(\mathsf{Q}(l|\cdot) + h) \colon h \in \mathcal{H}_l\}) &= \mathop{\mathbb{E}}_{\substack{S \sim \mathcal{D}^m \\ \boldsymbol{\sigma}}} \left[ \sup_{h \in \mathcal{H}_l} \sum_{i=1}^{m_k} \sigma_i [h(x_i) + \mathsf{Q}(l|x_i)] \right] \\
&= \mathop{\mathbb{E}}_{\substack{S \sim \mathcal{D}^m \\ \boldsymbol{\sigma}}} \left[ \sup_{h \in \mathcal{H}_l} \sum_{i=1}^{m_k} \sigma_i h(x_i) \right] + \mathop{\mathbb{E}}_{\substack{S \sim \mathcal{D}^m \\ \boldsymbol{\sigma}}} \left[ \sum_{i=1}^{m_k} \sigma_i \mathsf{Q}(l|x_i) \right] \\
&= \mathop{\mathbb{E}}_{\substack{S \sim \mathcal{D}^m \\ \boldsymbol{\sigma}}} \left[ \sup_{h \in \mathcal{H}_l} \sum_{i=1}^{m_k} \sigma_i h(x_i) \right] + \mathop{\mathbb{E}}_{S \sim \mathcal{D}^m} \left[ \sum_{i=1}^{m_k} \mathop{\mathbb{E}}_{\boldsymbol{\sigma}}[\sigma_i] \mathsf{Q}(l|x_i) \right] \\
&= \mathop{\mathbb{E}}_{\substack{S \sim \mathcal{D}^m \\ \boldsymbol{\sigma}}} \left[ \sup_{h \in \mathcal{H}_l} \sum_{i=1}^{m_k} \sigma_i h(x_i) \right] = \mathfrak{R}_{m_k}(\mathcal{H}_l).
\end{aligned}
$$

Similarly, we have $\mathfrak{R}_{m_k}(\{(\mathsf{Q}(l|\cdot) - h) \colon h \in \mathcal{H}_l\}) = \mathfrak{R}_{m_k}(\mathcal{H}_l)$. This completes the proof. $\qquad \square$

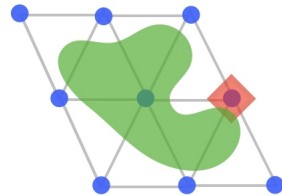

**Figure 4:** Illustration of the set $\overline{\Lambda}$, vertices of simplices(in blue). The area in green represents the set $\Lambda$. The small area in pink shows an $\epsilon$-ball in $l_1$-distance (and hence a lozenge) around a vertex.

**Theorem 4.** *For any $\delta > 0$, with probability at least $1 - \delta$ over the draw of a sample $S = (S_1, \dots, S_p) \sim \mathcal{D}_1^{m_1} \otimes \cdots \otimes \mathcal{D}_p^{m_p}$, the following inequality holds for all ensemble functions $f = \sum_{t=1}^{T} \alpha_t Q(k_t | \cdot) h_t \in \mathcal{F}$ and all $\rho \in (0,1)$ and $\lambda \in \Delta$:*

$$\mathcal{L}(\mathcal{D}_\lambda, f) \leq \mathcal{L}_\rho(\overline{\mathcal{D}}_\lambda, f) + \sum_{k=1}^{p} \lambda_k \left[ \frac{2}{\rho} \max_{l \in [p]} \mathfrak{R}_{m_k}(\mathcal{H}_l) + \sqrt{\frac{\log \log_2 \frac{2}{\rho}}{m_k}} + \sqrt{\frac{\log \frac{p}{\delta}}{2m_k}} \right]. \qquad (13)$$

*Proof.* By the uniform margin bound ([Mohri et al., 2018](#), Theorem 5.9), for any $k \in [p]$, with probability at least $1 - \delta$ the following inequality holds for all $f \in \mathcal{F}$ and $\rho \in (0,1]$:

$$\mathcal{L}(\mathcal{D}_k, f) \leq \mathcal{L}_\rho(\widehat{\mathcal{D}}_k, f) + \frac{2}{\rho} \mathfrak{R}_{m_k}(\mathcal{F}) + \sqrt{\frac{\log \log_2 \frac{2}{\rho}}{m_k}} + \sqrt{\frac{\log \frac{1}{\delta}}{2m_k}}.$$

The rest of the proof is similar to that of Theorem [1](#). $\qquad \square$

## F   Finer margin-based learning guarantees

In this section, we give finer margin-based generalization bounds for the family of ensemble $\mathcal{F}$. These learning bounds are particularly more relevant in the case where $\Lambda$ is a strict subset of the simplex $\Delta$. $\Lambda$ may be in fact a much smaller set, motivated by prior knowledge about the task and thus possible target mixtures. In some instances, it may even be a finite subset, which corresponds to only a finite set of mixtures.

For any family of real-valued functions $\mathcal{G}$, define the *weighted Rademacher complexity of $\mathcal{G}$* for the vector of samples $S_k = (z_{k,1}, \dots, z_{k,m_k})$ of sizes $\mathbf{m} = (m_1, \dots, m_p)$ as follows:

$$\mathfrak{R}_{\mathbf{m}}(\mathcal{G}, \lambda) = \mathbb{E}_{S_k \sim \mathcal{D}_k^{m_k}, \boldsymbol{\sigma}} \left[ \sup_{g \in \mathcal{G}} \sum_{k=1}^{p} \frac{\lambda_k}{m_k} \sum_{i=1}^{m_k} \sigma_{k,i} g(z_{k,i}) \right]. \qquad (14)$$

Fix $\lambda \in \Lambda$ and define $\Psi(S)$ for the vector of training samples $S = (S_1, \dots, S_p)$ as follows:

$$\Psi(S) = \sup_{h \in \mathcal{G}} \left\{ \mathbb{E}_{z \sim \mathcal{D}_\lambda}[g(z)] - \mathbb{E}_{z \sim \overline{\mathcal{D}}_\lambda}[g(z)] \right\},$$

where $\overline{\mathcal{D}}_\lambda = \sum_{k=1}^{p} \lambda_k \widehat{\mathcal{D}}_k$, with $\widehat{\mathcal{D}}_k$ the empirical distribution associated with the sample $S_k$. Assume that functions in $\mathcal{G}$ take values in $[0, 1]$. For any vector of samples $S'$ differing from $S$ only by point $z'_{k,i}$ in $S'_k$ and $z_{k,i}$ in $S_k$, we have

$$\Psi(S') - \Psi(S) \leq \sup_{g \in \mathcal{G}} \left\{ \left\{ \mathbb{E}_{z \sim \mathcal{D}_\lambda}[g(z)] - \mathbb{E}_{z \sim \overline{\mathcal{D}}'_\lambda}[g(z)] \right\} - \left\{ \mathbb{E}_{z \sim \mathcal{D}_\lambda}[g(z)] - \mathbb{E}_{z \sim \overline{\mathcal{D}}_\lambda}[g(z)] \right\} \right\}$$

$$= \sup_{g \in \mathcal{G}} \left\{ \mathbb{E}_{z \sim \overline{\mathcal{D}}_\lambda}[g(z)] - \mathbb{E}_{z \sim \overline{\mathcal{D}}'_\lambda}[g(z)] \right\} = \sup_{g \in \mathcal{G}} \frac{\lambda_k}{m_k} \left[ g(z_{k,i}) - g(z'_{k,i}) \right] \leq \frac{\lambda_k}{m_k}.$$

Furthermore, as with the standard Rademacher complexity ([Mohri et al., 2018](#)), the expectation can be upper bounded in terms of the weighted Rademacher complexity:

$\mathbb{E}_{S_k \sim \mathcal{D}_k^{m_k}} \left[ \sup_{g \in \mathcal{G}} \mathbb{E}_{z \sim \mathcal{D}_\lambda}[g(z)] - \mathbb{E}_{z \sim \overline{\mathcal{D}}_\lambda}[g(z)] \right] \leq 2\mathfrak{R}_{\mathbf{m}}(\mathcal{G}, \lambda)$. Thus, by McDiarmid's inequality, for any $\delta > 0$, with probability at least $1 - \delta$,

$$\mathbb{E}_{z \sim \mathcal{D}_\lambda}[g(z)] \leq \mathbb{E}_{z \sim \overline{\mathcal{D}}_\lambda}[g(z)] + 2\mathfrak{R}_{\mathbf{m}}(\mathcal{G}, \lambda) + \sqrt{\sum_{k=1}^{p} \frac{\lambda_k^2}{2m_k} \log \frac{1}{\delta}}.$$

Let $\overline{\Lambda}$ be the set of vertices of a subsimplicial cover of $\Lambda$, that is a decomposition of a cover of $\Lambda$ into subsimplices. When the subsimples are formed by vertices that are $\epsilon$-close in $\ell_1$-distance, then $\overline{\Lambda}$ is an $\epsilon$-cover of $\Lambda$ for the $\ell_1$-distance. Figure 4 illustrates the sets $\lambda$ and $\overline{\Lambda}$. By the union bound, with probability at least $1 - \delta$, the following holds for all $\lambda \in \overline{\Lambda}$:

$$\mathbb{E}_{z \sim \mathcal{D}_\lambda}[g(z)] \leq \mathbb{E}_{z \sim \overline{\mathcal{D}}_\lambda}[g(z)] + 2\mathfrak{R}_{\mathbf{m}}(\mathcal{G}, \lambda) + \sqrt{\sum_{k=1}^{p} \frac{\lambda_k^2}{2m_k} \log \frac{|\overline{\Lambda}|}{\delta}}.$$

Now, fix $\rho > 0$. Let $\mathcal{H}$ be a hypothesis set of real-valued functions and let $\phi_\rho$ denote the $\rho$-ramp loss. Let $\mathcal{G}$ be the family of $\rho$-ramp losses of functions in $\mathcal{H}$: $\mathcal{G} = \{z = (x, y) \mapsto \phi_\rho(yh(x)) : h \in \mathcal{H}\}$. Then, proceeding as with the derivation of margin-based Rademacher complexity bounds in the standard case and using the $\frac{1}{\rho}$-Lipschitzness of the $\rho$-ramp loss (Mohri et al., 2018), we obtained that, with probability at least $1 - \delta$, the following holds for all $\lambda \in \overline{\Lambda}$:

$$\mathcal{L}(\mathcal{D}_\lambda, h) \leq \mathcal{L}_\rho(\overline{\mathcal{D}}_\lambda, h) + \frac{2}{\rho}\mathfrak{R}_{\mathbf{m}}(\mathcal{H}, \lambda) + \sqrt{\sum_{k=1}^{p} \frac{\lambda_k^2}{2m_k} \log \frac{|\overline{\Lambda}|}{\delta}}.$$

Now, for any $\lambda, \lambda' \in \Delta$ with $\|\lambda - \lambda'\|_1 \leq \epsilon$, we have:

$$
\begin{aligned}
\sum_{k=1}^{p} \frac{\lambda_k'^2}{2m_k} &= \sum_{k=1}^{p} \frac{(\lambda_k' - \lambda_k + \lambda_k)^2}{2m_k} \\
&= \sum_{k=1}^{p} \frac{\lambda_k^2 + 2(\lambda_k' - \lambda_k)\lambda_k + (\lambda_k' - \lambda_k)^2}{2m_k} \\
&= \sum_{k=1}^{p} \frac{\lambda_k^2}{2m_k} + \sum_{k=1}^{p} \frac{2|\lambda_k' - \lambda_k|\lambda_k + (\lambda_k' - \lambda_k)^2}{2m_k} \\
&\leq \sum_{k=1}^{p} \frac{\lambda_k^2}{2m_k} + \epsilon \max_{k \in [p]} \frac{\lambda_k}{m_k} + \frac{\epsilon^2}{2} \max_{k \in [p]} \frac{1}{m_k} \qquad \text{(Hölder's inequality)} \\
&\leq \sum_{k=1}^{p} \frac{\lambda_k^2}{2m_k} + \frac{3\epsilon}{2}.
\end{aligned}
$$

Let $\overline{\Lambda}_\epsilon$ denote the family of $\lambda$s that are $\epsilon$-close to $\overline{\Lambda}$ in $\ell_1$-distance, then, for any $\lambda \in \overline{\Lambda}_\epsilon$ we have:

$$\mathcal{L}(\mathcal{D}_\lambda, h) \leq \mathcal{L}_\rho(\overline{\mathcal{D}}_\lambda, h) + \frac{2}{\rho}\mathfrak{R}_{\mathbf{m}}(\mathcal{H}, \lambda) + \frac{3\epsilon}{2} + \sqrt{\sum_{k=1}^{p} \frac{\lambda_k^2}{2m_k} \log \frac{|\overline{\Lambda}|}{\delta}}.$$

Also, for any $\lambda$ in a subsimplex formed by elements of $\overline{\Lambda}$, there exist $\mu = (\mu_1, \ldots, \mu_p)$ and $\beta_1, \ldots, \beta_p$ in $\overline{\Lambda}$ such that $\lambda = \sum_{k=1}^{p} \mu_k \beta_k$. Thus, for any such $\lambda$, we have

$$\mathcal{L}(\mathcal{D}_\lambda, h) \leq \mathcal{L}_\rho(\overline{\mathcal{D}}_\lambda, h) + \frac{2}{\rho} \sum_{l=1}^{p} \mu_k \mathfrak{R}_{\mathbf{m}}(\mathcal{H}, \beta_l) + \sum_{k=1}^{p} \mu_k \sqrt{\sum_{l=1}^{p} \frac{\beta_{l,k}^2}{2m_k} \log \frac{|\overline{\Lambda}|}{\delta}}.$$

Applying these results to the analysis of the Q-ensembles we are interested yields the following margin-based generalization bounds.

**Theorem 2.** *Fix $\rho > 0$ and $\epsilon > 0$. Then, for any $\delta > 0$, with probability at least $1 - \delta$ over the draw of a sample $S = (S_1, \ldots, S_p) \sim \mathcal{D}_1^{m_1} \otimes \cdots \otimes \mathcal{D}_p^{m_p}$, each of the following inequalities holds:*
*1. for all ensemble functions $f = \sum_{t=1}^{T} \alpha_t \mathsf{Q}(k_t|\cdot)h_t \in \mathcal{F}$ and all $\lambda \in \overline{\Lambda}_\epsilon$:*

$$\mathcal{L}(\mathcal{D}_\lambda, h) \leq \mathcal{L}_\rho(\overline{\mathcal{D}}_\lambda, h) + \frac{2}{\rho} \max_{r \in [p]} \mathfrak{R}_{\mathbf{m}}(\mathcal{H}_r, \lambda) + \frac{3\epsilon}{2} + \sqrt{\sum_{k=1}^{p} \frac{\lambda_k^2}{2m_k} \log \frac{|\overline{\Lambda}|}{\delta}}. \tag{6}$$

*2. for all ensemble functions $f = \sum_{t=1}^{T} \alpha_t \mathsf{Q}(k_t|\cdot)h_t \in \mathcal{F}$ and all $\lambda = \sum_{k=1}^{p} \mu_k \beta_k \in \mathrm{conv}(\overline{\Lambda})$:*

$$\mathcal{L}(\mathcal{D}_\lambda, h) \leq \mathcal{L}_\rho(\overline{\mathcal{D}}_\lambda, h) + \frac{2}{\rho} \sum_{l=1}^{p} \mu_k \max_{r \in [p]} \mathfrak{R}_{\mathbf{m}}(\mathcal{H}_r, \beta_l) + \sum_{k=1}^{p} \mu_k \sqrt{\sum_{l=1}^{p} \frac{\beta_{l,k}^2}{2m_k} \log \frac{|\overline{\Lambda}|}{\delta}}. \tag{7}$$

Note that, for a given $\lambda \in \Lambda$, the most favorable of the two statements of the theorem can be used. Observe also that the second learning bound coincides with that of Theorem 1 when $\overline{\Lambda}$ is chosen to be the vertices of the simplex $\Delta$ since in that case $|\overline{\Lambda}| = p$, $\mathrm{conv}(\overline{\Lambda}) = \Delta$, and since $\mathfrak{R}_{\mathbf{m}}(\mathcal{H}_r, \beta_l) = \mathfrak{R}_{\mathbf{m}}(\mathcal{H}_r, \beta_l)$ then coincides with $\mathfrak{R}_{m_l}(\mathcal{H}_r)$. Choosing the best statement of the theorem therefore always provides a finer guarantee than that of Theorem 1.

When $\Lambda$ is a small set, for example the set of $\lambda$s $\epsilon$-close to a finite set of discrete elements $\overline{\Lambda}$, then the last term of the learning bound of the first statement can be more favorable that of Theorem 1 since $|\overline{\Lambda}|$ can then be in the same order as $p$ or smaller while, by the sub-additivity of the square-root, the following inequality holds: $\sqrt{\sum_{k=1}^{p} \frac{\lambda_k^2}{m_k}} \leq \sum_{k=1}^{p} \sqrt{\frac{\lambda_k^2}{m_k}} = \sum_{k=1}^{p} \lambda_k \sqrt{\frac{1}{m_k}}$.

The theorem suggests a regularization term of the form $\sum_{k=1}^{p} \frac{\lambda_k^2}{m_k}$, especially in the case where $\Lambda$ is a small subset of the simplex, which can lead to better algorithms in that case.

## G  FEDMULTIBOOST: related work and experiments

Following MULTIBOOST, we propose a boosting-style approach with the agnostic loss. Boosting in federated learning was first studied by Hamer et al. (2020). The authors proposed a communication-efficient algorithm for minimizing the standard empirical risk and agnostic loss, based on mirror descent. However, their algorithm is optimal only for density estimation (Hamer et al., 2020, Section 3.2) and is sub-optimal for general classification tasks such as in Proposition 1. Furthermore, their mirror descent solution is inadequate for the boosting framework in this paper, where a (block) coordinate descent approach for learning sparser solutions is critical. Recently, Shen et al. (2021) proposed a functional gradient boosting algorithm for federated learning. Their algorithm iteratively determines base classifiers and mixing weights to compute a convex combination in a distributed manner. However, their algorithm minimizes the uniform loss over all samples. In contrast, we propose to minimize the agnostic loss over multiple domains, which is more risk-averse, and seek Q-ensembles which are more adequate than convex combinations in a multiple-source scenario.

**Federated learning experiments.** We used the *EMNIST* dataset (Caldas et al., 2018; Bonawitz et al., 2019), which contains 32x32 pixel handwritten digits images annotated by users. The images are divided into $p = 2$ sources based on the group of writers that provided the annotation: high school ($k = 1$) and census ($k = 2$). We compared algorithms on two binary classification problems: digits 4 vs. 9 and digits 1 vs. 7. The results are presented in Table 2. The error bars are obtained from breaking the set of clients into 10 random folds.

We compared FEDMULTIBOOST with three benchmarks, FEDADABOOST-k for $k \in [p]$ and FEDAD-ABOOST-all. The former is a federated version of ADABOOST algorithm trained only on a single source $k$ and the latter is federated ADABOOST trained on both the sources. In the federated AD-ABOOST versions, at each boosting step we randomly select 20 clients and train weak learners on each of those clients, next, the server selects the weak learner with the smallest weighted error and adds it to the ensemble. For the FEDMULTIBOOST algorithm, we randomly select 20 clients per round. Since the number of clients sampled at each round is small, we run each algorithm for 500 boosting steps.

As can be seen from Table 2, FEDMULTIBOOST provides agnostic and uniform errors that are significantly better than the baselines on both the datasets.

**Table 2:** Test errors for multiple benchmarks in the federated setting.

| Algorithm | Error-1 | Error-2 | Error-Uniform | Error-Agnostic |
|---|---|---|---|---|
| EMNIST (4 vs. 9) | | | | |
| FEDADABOOST-1 | $0.075 \pm 0.008$ | $0.133 \pm 0.014$ | $0.104 \pm 0.009$ | $0.133 \pm 0.014$ |
| FEDADABOOST-2 | $0.095 \pm 0.009$ | $0.096 \pm 0.014$ | $0.095 \pm 0.012$ | $0.096 \pm 0.014$ |
| FEDADABOOST-all | $0.076 \pm 0.006$ | $0.125 \pm 0.016$ | $0.101 \pm 0.011$ | $0.125 \pm 0.007$ |
| FEDMULTIBOOST | $\mathbf{0.064 \pm 0.013}$ | $\mathbf{0.076 \pm 0.008}$ | $\mathbf{0.070 \pm 0.009}$ | $\mathbf{0.076 \pm 0.016}$ |
| EMNIST (1 vs. 7) | | | | |
| FEDADABOOST-1 | $\mathbf{0.029 \pm 0.010}$ | $0.050 \pm 0.009$ | $0.039 \pm 0.011$ | $0.050 \pm 0.009$ |
| FEDADABOOST-2 | $0.062 \pm 0.014$ | $\mathbf{0.030 \pm 0.007}$ | $0.046 \pm 0.014$ | $0.062 \pm 0.007$ |
| FEDADABOOST-all | $0.032 \pm 0.007$ | $0.043 \pm 0.006$ | $0.037 \pm 0.007$ | $0.043 \pm 0.014$ |
| FEDMULTIBOOST | $0.030 \pm 0.008$ | $0.035 \pm 0.006$ | $\mathbf{0.032 \pm 0.008}$ | $\mathbf{0.035 \pm 0.010}$ |

# H Dataset references and details

- MNIST: `http://yann.lecun.com/exdb/mnist/`
- SVHN: `http://ufldl.stanford.edu/housenumbers/`
- MNIST-M: `http://yaroslav.ganin.net/`
- SENTIMENT: `https://www.cs.jhu.edu/~mdredze/datasets/sentiment`
- FASHION-MNIST: `https://github.com/zalandoresearch/fashion-mnist`
- ADULT: `https://archive.ics.uci.edu/ml/datasets/adult`

**Table 3:** Number of examples per domain for each classification problem.

| Problem | Source k=1 | Source k=2 | Source k=3 | Total |
|---|---|---|---|---|
| Sentiment Analysis | 2,000 | 2,000 | 2,000 | 6,000 |
| Digit Classification (1 vs 7) | 15,170 | 26,574 | 14,728 | 56,472 |
| Digit Classification (4 vs 9) | 13,782 | 16,235 | 13,406 | 43,423 |
| Object Recognition | 21,000 | 21,000 | 28,000 | 70,000 |
| Tabular Data (Adult Data) | 10,628 | 14,783 | 19,811 | 45,222 |

# I  Supplementary plots

This section contains additional plots illustrating the performance of the proposed MULTIBOOST algorithm. We illustrate convergence characteristics, Figure 7, $Q(k|\cdot_k)$ functions, Figure 5, and $\alpha$-mass distributions over the domains and $Q(k|\cdot_k)$ functions, Figure 6.

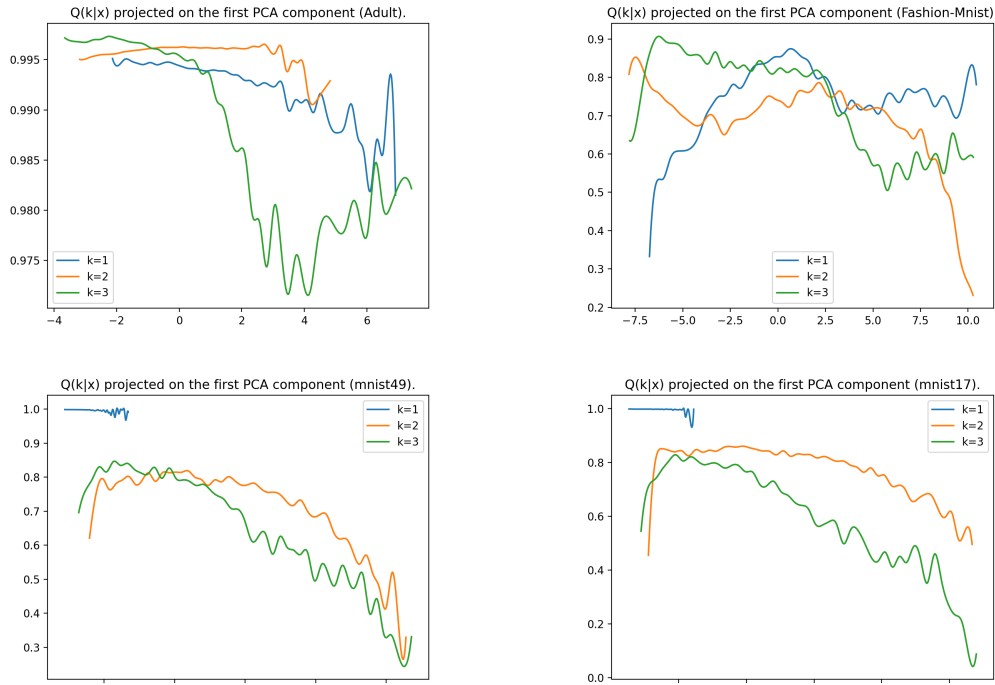

**Figure 5:** Mean values of $Q(k|\cdot_k)$ for the three domains of the data projected on the first principal component of the joint data.

**Top, left:** *Adult* data. Source $k = 1$ consists of individuals with a university degree (BSc, MS or PhD), source $k = 2$ those with only a High School diploma, and source $k = 3$, none of the above.

**Top, right:** *Fashion-MNIST* data. Source $k = 1$ consists of `t-shirts, pullovers, trousers`; $k = 2$ consists of `dresses, coats, sandals`; $k = 3$ consists of `shirts, bags, sneakers, ankle boots`.

**Bottom, left:** Digits (4 vs. 9), where pixel handwritten digits images are taken from sources: *MNIST* ($k = 1$), *SVHN* ($k = 2$), *MNIST-M* ($k = 3$).

**Bottom, right:** Digits (1 vs. 7), where pixel handwritten digits images are taken from sources: *MNIST* ($k = 1$), *SVHN* ($k = 2$), *MNIST-M* ($k = 3$).

Note that for the two bottom plots domain $k = 1$ is significantly further separated from the other two domains, since the pixels for $k = 1$ are ⸜⸜ ⸜⸜ ⸜⸜ and therefore $k = 2, 3$ ⸜⸜ ⸜⸜ le.

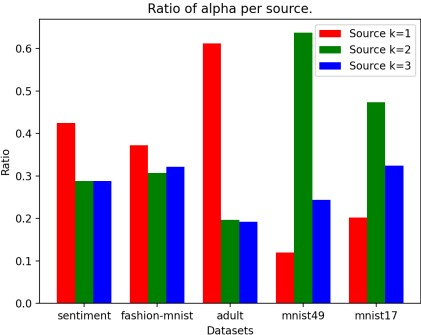

**Figure 6:** The ratio of ensemble weights $\alpha_{k,j}$ after training that corresponds to each source $k = 1, 2, 3$. For each dataset and fixed $k$, the bar corresponds to $\sum_{j=1}^{N_k} \alpha_{k,j} / \sum_{k=1}^{p} \sum_{j=1}^{N_k} \alpha_{k,j}$.

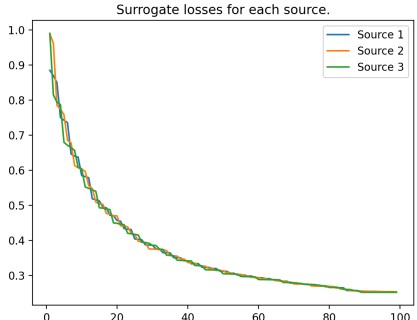 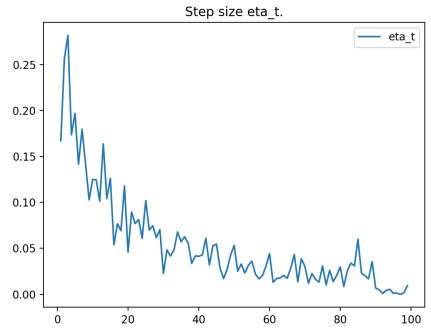

**Figure 7:** Left: The evolution of surrogate losses $\Phi$ for sources $k = 1, 2, 3$ during MULTIBOOST training on object recognition problem (*Fashion-MNIST*). Line $k$ corresponds to $\frac{1}{m_k} \sum_{i=1}^{m_k} \Phi(y_i, f_t(x_i))$ as a function of $t$, where $f_t$ is the MULTIBOOST ensemble of weak learners obtained at round $t$. Right: The evolution of the step size $\eta_t$ during MULTIBOOST training on object recognition problem (*Fashion-MNIST*).

## J    Comparison with multiple-source adaptation

In this section, we compare our results with an alternative solution based on using a multiple-source adaptation algorithm (Cortes, Mohri, Suresh, and Zhang, 2021; Mansour, Mohri, and Rostamizadeh, 2008, 2009a; Hoffman, Mohri, and Zhang, 2018, 2021). Let us emphasize first that these algorithms are designed for a distinct scenario than the one studied in this paper. They assume no access to labeled data and instead only to a good predictor and unlabeled data for each domain.

The method consists of first training AdaBoost on each domain, which provides an accurate predictor $f_k$ for each domain $k$. Next, we can use the discriminative technique for multiple-source adaptation (DMSA algorithm) recently presented by Cortes, Mohri, Suresh, and Zhang (2021) to combine these predictors to derive a solution that is robust for any target mixture distribution of the source domains. This algorithm was shown to outperform the GMSA algorithm of Hoffman, Mohri, and Zhang (2018) and Hoffman, Mohri, and Zhang (2021), which is based on density estimation (note that previous work by Mansour, Mohri, and Rostamizadeh (2008, 2009a) did not provide an actual algorithm for this problem), which itself was shown to surpass other existing algorithms for this scenario.

As mentioned earlier, the idea of Q-ensembles in our boosting context is inspired by the distribution-weighted combinations of Mansour, Mohri, and Rostamizadeh (2008, 2009a) or Hoffman, Mohri, and Zhang (2018, 2021) or the domain-classifier based combinations of Cortes, Mohri, Suresh, and Zhang (2021). Our basic motivation via Propositions 1 and 2 are also similar. However, the main technical content of this work and the contributions are all novel. Our formulation of the multiple-source boosting problem, including our weak learning assumption, our algorithmic solutions, including an extension to the Federated Learning setting, and our theoretical analysis of the problem, including finer margin-based learning bounds, our extensive experimental results, are all novel and unrelated to that previous work.

In Section 6, we report experimental results for this AdaBoost and DMSA-based algorithm, and compare them with MULTIBOOST. Note that, as already mentioned, DMSA is designed for a different scenario from the one studied in this paper where no access to labeled data is assumed. The empirical results suggest that MULTIBOOST outperforms DMSA, although DMSA often achieves a competitive performance in the tasks examined.

Let us emphasize, however, that the predictor for the DMSA algorithm (or GMSA) in general does not benefit from informative guarantees in our scenario, for the following reasons.

**Target labeling function assumption**: the main analysis and results in (Mansour, Mohri, and Rostamizadeh, 2008, 2009a; Hoffman, Mohri, and Zhang, 2018, 2021; Cortes, Mohri, Suresh, and Zhang, 2021) require that the target labeling function (or conditional probability of $Y$ given $X$ for an extension of that analysis) be the same for all domains. This is a strong condition that may not hold in practice and that is not required for our learning guarantees for MULTIBOOST.

**Loss function**: the guarantees in (Mansour, Mohri, and Rostamizadeh, 2008, 2009a; Hoffman, Mohri, and Zhang, 2018, 2021; Cortes, Mohri, Suresh, and Zhang, 2021) hold only for a continuous loss function, since they rely on Brouwer's fixed-point theorem. In particular, they do not hold for the binary or multi-class mis-classification losses considered here.

One can resort instead to a convex surrogate loss (such a guarantee would be then in terms of the convex loss of the predictors to combine and not their more favorable zero-one loss). But the guarantees in (Mansour, Mohri, and Rostamizadeh, 2008, 2009a; Hoffman, Mohri, and Zhang, 2018, 2021; Cortes, Mohri, Suresh, and Zhang, 2021) also require the loss to be bounded, which would not hold for an unbounded domain. Even for a bounded domain, the bound could be large and the value of the convex loss on the boundary even larger (exponentially larger for AdaBoost) thereby making the desired guarantee essentially vacuous. In contrast, our guarantees hold for the zero-one loss.

**Algorithms**: the technique of Mansour, Mohri, and Rostamizadeh (2008, 2009a) and Hoffman, Mohri, and Zhang (2018, 2021) requires density estimation for each domain. With estimated densities, the guarantee becomes somewhat looser. Furthermore, the algorithmic solutions of (Hoffman, Mohri, and Zhang, 2018, 2021; Cortes, Mohri, Suresh, and Zhang, 2021) are not based on a convex optimization and thus cannot directly benefit from the theoretical bound, even if it could be applicable (see above).

**Hypothesis set**: ignoring the normalization, which does not affect the definition of the classifier, the DMSA solution is a specific element of the family of ensembles our algorithm and learning bounds consider. Our algorithm seeks the most favorable ensemble using all the available labeled data, which in general could be quite different and more favorable than the predictor obtained by combining Adaboost and DMSA.

More generally, our algorithm is not restricted to deriving intermediate predictors for each domain and instead directly exploits the labeled training samples from all the sources simultaneously to find a single good predictor. Consider the extreme case where all sources follow the same distribution and all training sets have the same size. The AdaBoost and DMSA-based algorithm consists of first training one AdaBoost model for each source. Assuming perfect density estimation or domain classification, DMSA would then return the uniform average $\frac{1}{p} \sum_{k=1}^{p} f_k$ with each $f_k$ trained on $m/p$ samples. Instead, MULTIBOOST returns a single model trained on all $m$ samples that, in general, can be far superior. To further illustrate this, we ran this experiment on the SVHN dataset with 3 *sources* defined by sub-samples from an original training sample. This led to an error of $26.1\%$ for the predictor obtained via DMSA and only $22.8\%$ for MULTIBOOST.

# K  The impact of domain classifier Q

We here present results illustrating the importance of selecting a high-accuracy domain classifier Q in the MULTIBOOST algorithm. We experimented with different Q functions by varying the number of steps, $C = 2, 5, 10, 1000$, in training the logistic regression optimizer used to obtain Q. The lower the number of steps in the logistic regression optimizer, the lower the quality of the domain classifier Q, and thus the higher the classification error on the underlying task. In the original MULTIBOOST implementation, the maximum number of L-BFGS steps for the domain classifier Q is set to 1000 by default.

**Table 4:** Test errors for original MULTIBOOST and MULTIBOOST-$C$, where $C$ is the maximum number of steps in the L-BFGS optimizer used to fit the multinomial logistic regression as domain classifier Q. In the original MULTIBOOST implementation, the maximum number of L-BFGS steps for the domain classifier Q is set to 1000 by default.

| Algorithm-C | Error-1 | Error-2 | Error-3 | Error-Uniform | Error-Agnostic |
|---|---|---|---|---|---|
| | | Digits Recognition (1 vs. 7) | | | |
| MULTIBOOST | **0.026 ± 0.004** | **0.261 ± 0.013** | **0.257 ± 0.015** | **0.181 ± 0.005** | **0.261 ± 0.011** |
| MULTIBOOST-10 | 0.029 ± 0.005 | 0.262 ± 0.013 | 0.299 ± 0.015 | 0.197 ± 0.010 | 0.299 ± 0.011 |
| MULTIBOOST-5 | 0.032 ± 0.005 | 0.277 ± 0.012 | 0.323 ± 0.017 | 0.211 ± 0.012 | 0.323 ± 0.015 |
| MULTIBOOST-2 | 0.127 ± 0.009 | 0.281 ± 0.015 | 0.351 ± 0.021 | 0.253 ± 0.017 | 0.351 ± 0.019 |

## L   Multi-Class extension of MULTIBOOST

Here, we briefly describe the extension of MULTIBOOST to the multi-class setting, MCMULTIBOOST

We denote by $\mathcal{Y}$ the set of output labels or categories. The label associated by a hypothesis $f \colon \mathcal{X} \times \mathcal{Y} \to \mathbb{R}$ to input $x \in \mathcal{X}$ is given by $\operatorname{argmax}_{y \in \mathcal{Y}} f(x,y)$. The margin $\rho_f(x,y)$ of the function $f$ for a labeled example $(x,y) \in \mathcal{X} \times \mathcal{Y}$ is defined by

$$\rho_f(x,y) = f(x,y) - \max_{y' \neq y} f(x,y'). \tag{15}$$

Thus, $f$ misclassifies $(x,y)$ when $\rho_f(x,y) \leq 0$.

As in the binary classification case, we consider a scenario where the learner receives labeled samples from $p$ source domains, each defined by a distribution $\mathcal{D}_k$ over $\mathcal{X} \times \mathcal{Y}$, $k \in [p]$. We denote by $S_k = ((x_{k,1}, y_{k,1}), \ldots, (x_{k,m_k}, y_{k,m_k})) \in (\mathcal{X} \times \mathcal{Y})^{m_k}$ the labeled sample of size $m_k$ received from source $k$, which is drawn i.i.d. from $\mathcal{D}_k$. For any function $f \colon \mathcal{X} \times \mathcal{Y} \to \mathbb{R}$ and distribution $\mathcal{D}$ over $\mathcal{X} \times \mathcal{Y}$, let $\mathcal{L}(\mathcal{D}, f)$ be the expected loss of $f$, that is $\mathcal{L}(\mathcal{D}, f) = \mathbb{E}_{(x,y) \sim \mathcal{D}}[\ell(f, x, y)]$, where $\ell$ is the multi-class loss $\ell(f(x), y) = \mathbb{I}(\rho_f(x,y) \leq 0)$.

For any $k \in [p]$, let $\mathcal{H}_k$ denote a hypothesis set of functions mapping from $\mathcal{X} \times \mathcal{Y}$ to the interval $[-1, +1]$, $|\mathcal{H}_k| = N_k$. The objective of the learner is to find an accurate predictor $f$ for any target distribution $\mathcal{D}_\lambda$ that is a mixture of the source distributions, where $\lambda$ may be in a subset $\Lambda$ of the simplex.

### L.1   Form of solution

In the multi-class setting, the general form of our Q-ensemble predictor is the following:

$$\forall (x,y) \in \mathcal{X} \times \mathcal{Y}, \ f(x,y) = \sum_{l=1}^{p} \sum_{j=1}^{N_l} \alpha_{l,j} \mathsf{Q}(l|x) h_{l,j}(x,y), \tag{16}$$

where $h_{l,j} \in \mathcal{H}_l$, $\alpha_{l,j} \geq 0$ and $\sum_{j=1}^{N_l} \alpha_{l,j} = 1$ and where $\mathsf{Q}(l|x)$ denotes the conditional probability of domain $l$ given $x$.

### L.2   Weak learning assumption

As in the binary classification scenario, we will adopt a weak-learning assumption. Unlike the standard single source, our assumption here must hold for each source domain: for each domain $k \in [p]$ and any distribution $\mathsf{D}$ over $S_k \times \mathcal{Y}$, there exists a base classifier $h \in \mathcal{H}_k$ such that the weighted loss of $h$ is $\gamma$-better than random: $\frac{1}{2}[1 - \mathbb{E}_{(i,y) \sim \mathsf{D}}[h(x_{k,i}, y_{k,i}) - h(x_{k,i}, y)] \leq \frac{1}{2} - \gamma$, for some edge value $\gamma > 0$. This is equivalent to the existence of a weak-learning for a each domain, which is a mild assumption. As in the standard boosting scenario, this suggests that there exists a good *rule of thumb* for each domain. The key difference from the standard learning scenario, however, is that here we seek a Q-ensemble and further require it to be accurate for any target mixture $\mathcal{D}_\lambda$, $\lambda \in \Lambda$. In the next subsection, we present an algorithm, MCMULTIBOOST, for deriving an accurate Q-ensemble for any target mixture domain that belongs to the convex combination of the source domains.

### L.3   Algorithm

Let $\Phi$ be a convex, increasing and differentiable function such that $u \mapsto \Phi(-u)$ upper-bounds the binary loss $u \mapsto 1_{u \leq 0}$. $\Phi$ could be the exponential function as in ADABOOST or the logistic function, as in logistic regression. Using $\Phi$ to upper-bound the agnostic loss leads to the following objective function for an ensemble $f$ defined by (16) for any $\alpha = (\alpha_{l,j})_{(l,j) \in [p] \times [N_l]} \geq 0$:

$$F(\alpha) = \max_{\lambda \in \Lambda} \sum_{k=1}^{p} \frac{\lambda_k}{m_k} \sum_{i=1}^{m_k} \sum_{y \in \mathcal{Y}} \Phi\left(-\sum_{l=1}^{p} \sum_{j=1}^{N_l} \alpha_{l,j} \mathsf{Q}(l|x_{k,i}) \mathsf{h}_{l,j}(x_{k,i}, y_{k,i}, y)\right), \tag{17}$$

where $\mathsf{h}_{l,j}(x_{k,i}, y_{k,i}, y) = h_{l,j}(x_{k,i}, y_{k,i}) - h_{l,j}(x_{k,i}, y)$.

Here, we will consider the case where the set $\Lambda$ coincides with the full simplex, that is $\Lambda = \Delta$. $F$ can then be expressed more straightforwardly as $F = \max_{k=1}^p F_k$, with $F_k(\alpha) = \frac{1}{m_k} \sum_{i=1}^{m_k} \sum_{y \in \mathcal{Y}} \Phi\left(-\sum_{l=1}^p \sum_{j=1}^{N_l} \alpha_{l,j} \mathsf{Q}(l|x_{k,i}) \mathsf{h}_{l,j}(x_{k,i}, y_{k,i}, y)\right)$.

Since a convex function composed with an affine function is convex and a sum of convex functions is convex, $F$ is convex as the maximum of a set of convex functions. While convex, $F$ is not differentiable and, in general, coordinate descent may not succeed in such cases (Tseng, 2001; Luo and Tseng, 1992). This is because the algorithm may be *stuck* at a point where no progress is possible along any of the axis directions, while there exists a favorable descent along some other direction. However, we will show that, under the weak-learning assumption we adopted, at any point $\alpha$ and for each active function $F_k$, that is $F_k$ such that $F_k(\alpha) = F(\alpha)$, there exists a coordinate direction along which a descent progress is possible for each $F_k$. We will assume that these directions are also descent directions for $F$. More generally, it suffices in fact that one such direction admits this guarantee. Alternatively, as in the binary classification setting, one can replace the maximum with a *soft-max*, that is the $(x_1, \ldots, x_k) \mapsto \frac{1}{\mu} \log(\sum_{k=1}^p e^{\mu x_k})$ for $\mu > 0$, which leads to a continuously differentiable function with a close approximation for $\mu$ sufficiently large.

**Description**. Let $\alpha_{t-1}$ denote the value of the parameter vector $\alpha = (\alpha_{l,j})$ at the end of the $(t-1)$th iteration and let $\mathsf{f}_{t-1}$ be defined by $\mathsf{f}_{t-1} = \sum_{l=1}^p \sum_{j=1}^{N_l} \alpha_{t-1,l,j} \mathsf{Q}(l|\cdot) \mathsf{h}_{l,j}$. Coordinate descent at iteration $t$ consists of choosing a direction $\mathbf{e}_{q,r}$ corresponding to base classifier $h_{q,r}$ and a step value $\eta > 0$ to minimize $F(\alpha_{t-1} + \eta \, \mathbf{e}_{q,r})$. To select a direction, we consider the subdifferential of $F$ along any $\mathbf{e}_{q,r}$. Since functions $F_k$ are differentiable, by the subdifferential calculus for the maximum of functions, the subdifferential of $F$ at $\alpha_{t-1}$ along the direction $\mathbf{e}_{q,r}$ is given by:

$$\partial F(\alpha_{t-1}, \mathbf{e}_{q,r}) = \mathrm{conv}\{F_k'(\alpha_{t-1}, \mathbf{e}_{q,r}) \colon k \in \mathcal{K}_t\},$$

where $F_k'(\alpha_{t-1}, \mathbf{e}_{q,r})$ is the directional derivative of $F_k$ at $\alpha_{t-1}$ along the direction $\mathbf{e}_{q,r}$ and where $\mathcal{K}_t = \{k \in [p] \colon F_k(\alpha_{t-1}) = F(\alpha_{t-1})\}$. We will therefore consider the direction $\mathbf{e}_{q,r}$ with the largest absolute directional derivative value $|F_k'(\alpha_{t-1}, \mathbf{e}_{q,r})|$, $k \in \mathcal{K}_t$, but will restrict ourselves to $q = k$ since, as we shall see, that will be sufficient to guarantee a non-zero directional gradient. To do so, we will express $F_k'(\alpha_{t-1}, \mathbf{e}_{k,r})$ in terms of the distribution $\mathsf{D}_t(k, \cdot, \cdot)$ over $S_k \times \mathcal{Y}$ defined by $\mathsf{D}_t(k, i, y) = \frac{\Phi'(-\mathsf{f}_{t-1}(x_{k,i}, y_{k,i}, y))}{Z_{t,k}}$, with $Z_{t,k} = \sum_{i=1}^{m_k} \sum_{y \in \mathcal{Y}} \Phi'(-\mathsf{f}_{t-1}(x_{k,i}, y_{k,i}, y))$, for all $i \in [m_k]$ and $y \in \mathcal{Y}$:

$$F_k'(\alpha_{t-1}, h_{k,r}) = \frac{1}{m_k} \sum_{i=1}^{m_k} \sum_{y \in \mathcal{Y}} -\mathsf{Q}(k|x_{k,i}) \mathsf{h}_{k,r}(x_{k,i}, y_{k,i}, y) \Phi'\left(-\mathsf{f}_{t-1}(x_{k,i}, y_{k,i}, y)\right)$$

$$= \frac{Z_{t,k}}{m_k}[2\epsilon_{t,k,r} - 1],$$

where $\epsilon_{t,k,r} = \frac{1}{2}\left[1 - \mathbb{E}_{(i,y) \sim \mathsf{D}_t(k,\cdot,\cdot)}[\mathsf{Q}(k|x_{k,i}) \mathsf{h}_{k,r}(x_{k,i}, y_{k,i}, y)]\right]$ denotes the *weighted error* of $\mathsf{Q}(k|\cdot)h_{k,r}$. For any $s \in [m_k]$, since $x_{k,s}$ is a sample drawn from $\mathcal{D}_k$, we have $\mathsf{Q}(k|x_{k,s}) > 0$ and therefore we have: $\overline{\mathsf{Q}}_{t,k} = \sum_{s=1}^{m_k} \sum_{y \in \mathcal{Y}} \mathsf{D}_t(k, s, y) \mathsf{Q}(k|x_{k,s}) > 0$. Thus, we can write $\mathbb{E}_{(i,y) \sim \mathsf{D}_t(k,\cdot,\cdot)}[\mathsf{Q}(k|x_{k,i}) \mathsf{h}_{k,r}(x_{k,i}, y_{k,i}, y)]$ as

$$\sum_{i=1}^{m_k} \sum_{y \in \mathcal{Y}} \frac{\mathsf{D}_t(k, i, y) \mathsf{Q}(k|x_{k,i}) \mathsf{h}_{k,r}(x_{k,i}, y_{k,i}, y)}{\overline{\mathsf{Q}}_{t,k}} \overline{\mathsf{Q}}_{t,k}, = \mathop{\mathbb{E}}_{(i,y) \sim \mathsf{D}_t'(k,\cdot,\cdot)}[\mathsf{h}_{k,r}(x_{k,i}, y_{k,i}, y)] \overline{\mathsf{Q}}_{t,k},$$

where $\mathsf{D}_t'(k, i, y) = \frac{\mathsf{D}_t(k,i,y) \mathsf{Q}(k|x_{k,i})}{\overline{\mathsf{Q}}_{t,k}}$. By our weak-learning assumption, there exists $r \in N_k$ such that $\mathbb{E}_{(i,y) \sim \mathsf{D}_t'(k,\cdot,\cdot)}[\mathsf{h}_{k,r}(x_{k,i}, y_{k,i}, y)] \geq \gamma > 0$. For that choice of $r$, we have $\epsilon_{t,k,r} < \frac{1}{2} - \overline{\gamma}$, with $\overline{\gamma} = \gamma \overline{\mathsf{Q}}_{t,k} > 0$. In view of that, it suffices for us to search along the directions $h_{k,r}$ and we do not need to consider the directional derivative of $F_k$ along directions $h_{q,r}$ with $q \neq k$.

The direction chosen by our coordinate descent algorithm is thus defined by: $\mathrm{argmax}_{k \in \mathcal{K}_t, r \in [N_k]} \frac{Z_{t,k}}{m_k}[1 - 2\epsilon_{t,k,r}]$. Given the direction $\mathbf{e}_{k,r}$, the optimal step value $\eta$ is $\mathrm{argmin}_{\eta > 0} F(\alpha_{t-1} + \eta \mathbf{e}_{k,r})$. The pseudocode of our algorithm, MCMULTIBOOST, is provided in Figure 8. In the most general case, $\eta$ can be found via a line search or other numerical methods.

**Step size**. In some special cases, the line search can be executed using a simpler expression by using an upper bound on $F(\alpha_{t-1} + \eta \mathbf{e}_{k,r})$. Denoting $z_{l,i} = (x_{l,i}, y_{l,i}, y)$ and using the convexity of $\Phi$, since $\mathsf{Q}(k|x_{l,i})\mathsf{h}_{k,r}(z_{l,i}) = \frac{1+\mathsf{Q}(k|x_{l,i})\mathsf{h}_{k,r}(z_{l,i})}{2} \cdot (+1) + \frac{1-\mathsf{Q}(k|x_{l,i})\mathsf{h}_{k,r}(z_{l,i})}{2} \cdot (-1)$, the following holds for all $\eta \in \mathbb{R}$:

$$\Phi(-\mathsf{f}_{t-1}(z_{l,i}) - \eta \mathsf{Q}(k|x_{l,i})\mathsf{h}_{k,r}(z_{l,i})) \leq \frac{1+\mathsf{Q}(k|x_{l,i})\mathsf{h}_{k,r}(z_{l,i})}{2}\Phi(-\mathsf{f}_{t-1}(z_{l,i}) - \eta)$$
$$+ \frac{1-\mathsf{Q}(k|x_{l,i})\mathsf{h}_{k,r}(z_{l,i})}{2}\Phi(-\mathsf{f}_{t-1}(z_{l,i}) + \eta).$$

In the case of exponential and logistic functions, the following upper bounds can then be derived.

**Lemma 2.** *For any $k \in [p]$, the following upper bound holds when $\Phi$ is the exponential or the logistic function:*

$$F(\alpha_{t-1} + \eta \mathbf{e}_{k,r}) \leq \max_{l \in [p]} \frac{Z_{t,l}}{m_l}\big[(1 - \epsilon_{t,l,k,r})e^{-\eta} + \epsilon_{t,l,k,r}e^{\eta}\big],$$

*where $\epsilon_{t,l,k,r} = \frac{1}{2}\big[1 - \mathbb{E}_{(i,y)\sim \mathsf{D}_t(l,\cdot,\cdot)}[\mathsf{Q}(k|x_{l,i})\mathsf{h}_{k,r}(x_{l,i}, y_{l,i}, y)]\big]$.*

For any $k$, function $\eta \mapsto (1 - \epsilon_{t,l,k,r})e^{-\eta} + \epsilon_{t,l,k,r}e^{\eta}$ reaches its minimum for $\eta = \frac{1}{2}\log\frac{1-\epsilon_{t,l,k,r}}{\epsilon_{t,l,k,r}}$. When the maximum is achieved for $l = k$, the solution coincides with the familiar expression of the step size $\eta_t = \frac{1}{2}\log\frac{1-\epsilon_{t,k,r}}{\epsilon_{t,k,r}}$ used in ADABOOST.

**Q-function**. As discussed in the binary classification setting, the conditional probability functions $\mathsf{Q}(k|\cdot)$ are crucial to the definition of our algorithm. As pointed out earlier, Q-ensembles can help achieve accurate solutions in some cases that cannot be realized using convex combinations. Furthermore, for any $k \in [p]$, since $\mathsf{D}_t(k, \cdot, \cdot)$ is a distribution over the $S_k \times \mathcal{Y}$, it is natural to assume that for any $j \neq k$ we have $\mathbb{E}_{(s,y)\sim\mathsf{D}_t(k,\cdot)}[\mathsf{Q}(k|x_{k,s})] \geq \mathbb{E}_{(s,y)\sim\mathsf{D}_t(k,\cdot)}[\mathsf{Q}(j|x_{k,s})]$. This implies the following lower bound: $\mathbb{E}_{(s,y)\sim\mathsf{D}_t(k,\cdot)}[\mathsf{Q}(k|x_{k,s})] \geq \frac{1}{p}$, which in turn implies $\overline{\gamma} \geq \frac{\gamma}{p}$, since for any $x \in \mathcal{X}$, $\sum_{j=1}^{p}\mathsf{Q}(j|x) = 1$. In the special case where all domains coincide, we have $\mathsf{Q}(k|x_{k,s}) = \frac{1}{p}$ for all $s$ and this lower bound is reached. At another extreme, when all domains admit distinct supports, we have $\mathsf{Q}(k|x_{k,s}) = 1$ for all $s \in [m_k]$ and thus $\overline{\gamma} = \gamma$.

---

MCMULTIBOOST$(S_1, \ldots, S_p)$

1    $\alpha_0 \leftarrow 0$
2    **for** $t \leftarrow 1$ **to** $T$ **do**
3       $\mathsf{f}_{t-1} \leftarrow \sum_{l=1}^{p}\sum_{j=1}^{N_l}\alpha_{t-1,l,j}\mathsf{Q}(l|\cdot)\mathsf{h}_{l,j}$
4       $\tilde{\Phi}_k \leftarrow \frac{1}{m_k}\sum_{i=1}^{m_k}\sum_{y\in\mathcal{Y}}\Phi(-\mathsf{f}_{t-1}(x_{k,i}, y_{k,i}, y))$
5       $\mathcal{K}_t \leftarrow \left\{k : k \in \text{argmax}_{k\in[p]}\tilde{\Phi}_k\right\}$
6       **for** $k \in \mathcal{K}_t$ **do**
7         $Z_{t,k} \leftarrow \sum_{i=1}^{m_k}\sum_{y\in\mathcal{Y}}\Phi'(-\mathsf{f}_{t-1}(x_{k,i}, y_{k,i}, y))$
8         **for** $i \leftarrow 1$ **to** $m_k, y \in \mathcal{Y}$ **do**
9           $\mathsf{D}_t(k, i, y) \leftarrow \frac{\Phi'(-\mathsf{f}_{t-1}(x_{k,i}, y_{k,i}, y))}{Z_{t,k}}$
10      $(k, r) \leftarrow \text{argmax}_{k\in\mathcal{K}_t, r\in[N_k]} \frac{Z_{t,k}}{m_k}[1 - 2\epsilon_{t,k,r}]$
11      $\eta_t \leftarrow \text{argmin}_{\eta>0} F(\alpha_{t-1} + \eta\mathbf{e}_{k,r})$
12      $\alpha_t \leftarrow \alpha_{t-1} + \eta_t\mathbf{e}_{k,r}$
13    $f \leftarrow \sum_{l=1}^{p}\sum_{j=1}^{N_l}\alpha_{T,l,j}\mathsf{Q}(l|\cdot)h_{l,j}$
14    **return** $f$

---

**Figure 8:** Pseudocode of the MCMULTIBOOST algorithm. $\epsilon_{t,k,r} = \frac{1}{2}\big[1 - \mathbb{E}_{(i,y)\sim\mathsf{D}_t(k,\cdot,\cdot)}[\mathsf{Q}(k|x_{k,i})\mathsf{h}_{k,r}(x_{k,i}, y_{k,i}, y)]\big]$ denotes the weighted error of $\mathsf{Q}(k|\cdot)h_{k,r}$.

In practice, we do not have access to the true conditional probabilities $\mathsf{Q}(k|\cdot)$. Instead, as in the binary classification setting, we can derive accurate estimates $\widehat{\mathsf{Q}}(k|\cdot)$ using large unlabeled samples from the source domains, the *label* used for training being simply the domain index. This can be done using algorithms such as conditional maximum entropy models (Berger et al., 1996) (or multinomial logistic regression), which benefit from strong theoretical guarantees (Mohri et al., 2018, Chapter 13), or other rich models based on neural networks.

**Other variants of MCMULTIBOOST.** Other variants of MCMULTIBOOST, such as the one where, instead of the maximum, the *softmax* function or the sum is used can be defined and analyzed as in the binary setting (Appendix D).

**Table 5:** Test errors for multi-class problems with RandomForest.

| Algorithm | Error-1 | Error-2 | Error-3 | Error-Uniform | Error-Agnostic |
|---|---|---|---|---|---|
| | | | Fashion-MNIST | | |
| ADABOOST.MR-1 | **0.131 ± 0.019** | 0.185 ± 0.008 | 0.193 ± 0.017 | 0.169 ± 0.017 | 0.193 ± 0.019 |
| ADABOOST.MR-2 | 0.157 ± 0.020 | **0.152 ± 0.009** | 0.159 ± 0.023 | 0.156 ± 0.019 | 0.159 ± 0.019 |
| ADABOOST.MR-3 | 0.158 ± 0.015 | 0.173 ± 0.008 | **0.155 ± 0.015** | 0.162 ± 0.016 | 0.173 ± 0.018 |
| ADABOOST.MR-all | 0.141 ± 0.020 | 0.166 ± 0.011 | 0.163 ± 0.022 | 0.156 ± 0.021 | 0.166 ± 0.016 |
| MCMULTIBOOST | 0.132 ± 0.027 | 0.161 ± 0.018 | 0.155 ± 0.027 | **0.149 ± 0.027** | **0.161 ± 0.024** |

**Table 6:** Test errors for multi-class problems with CNNs.

| Algorithm | Error-1 | Error-2 | Error-3 | Error-Uniform | Error-Agnostic |
|---|---|---|---|---|---|
| ADABOOST.MR-1 | **0.083 ± 0.007** | 0.098 ± 0.006 | 0.104 ± 0.006 | 0.095 ± 0.006 | 0.104 ± 0.006 |
| ADABOOST.MR-2 | 0.104 ± 0.007 | **0.090 ± 0.003** | 0.099 ± 0.006 | 0.097 ± 0.003 | 0.104 ± 0.004 |
| ADABOOST.MR-3 | 0.098 ± 0.005 | 0.106 ± 0.003 | **0.092 ± 0.003** | 0.099 ± 0.005 | 0.106 ± 0.004 |
| ADABOOST.MR-all | 0.093 ± 0.005 | 0.096 ± 0.007 | 0.098 ± 0.007 | 0.095 ± 0.004 | 0.098 ± 0.007 |
| MCMULTIBOOST | 0.086 ± 0.006 | 0.090 ± 0.003 | 0.093 ± 0.005 | **0.089 ± 0.003** | **0.091 ± 0.006** |

## L.4 Experiments

Here, we report the results of several experiments with the MCMULTIBOOST algorithm on a multiple-source dataset with multiple classes. We present two sets of experiments: one where we used as base predictors random forest classifiers $\mathcal{H}^{RF}$, and another one where we used convolutional neural networks (CNNs).

We use multi-source data based on images of clothes items from the *Fashion-MNIST* (Xiao et al., 2017) dataset. We defined 3 domains: the first domain $(k = 1)$ coincides with the original Fashion-MNIST; the second and third domains $(k = 2, 3)$ are both defined as in Fashion-MNIST but with additive noise, with a different type of noise for $k = 2$ and $k = 3$.

As in Section 6, the probabilities $Q(k|\cdot)$, $k \in [p]$, are estimated using multinomial logistic regression (or conditional maximum entropy models). We compared MCMULTIBOOST with a set of multi-class extensions of ADABOOST that operate on the same hypotheses class $\mathcal{H}^{RF}$, which include ADABOOST-k for $k \in [p]$ and ADABOOST-all.

Tables 5 and 6 report our empirical results. The results show that, as in the binary classification setting, the extension of our algorithm to the multi-class setting, MCMULTIBOOST, outperforms all baselines for both sets of experiments (using Random forests or CNNs).