# OpenReview forum: "Boosting with Multiple Sources"
_NeurIPS.cc/2021/Conference — NeurIPS 2021 Poster_

### Official Review · Reviewer_Nwre · 2021-07-15

**Rating:** 6
**Confidence:** 4

**Summary:**

The paper studies the problem of learning boosted classifiers with data from multiple source domains. The authors propose Q-ensembles to deal with it and seek a solution for any mixture of the source distribution. They propose the algorithm MultiBoost to solve it inspired by coordinate descent. They present theoretical analysis in terms of margin-based learning bounds. Moreover, they present an extension to the federated learning scenario. In experiments, they compare experimental results with AdaBoost.

**Limitations And Societal Impact:**

The authors adequately addressed the limitations. The authors consider a more practical scene, federated learning scenario. Their FedMultiBoost algorithm, to some extent, makes the solution fair and risk-averse.

**Main Review:**

-	Originality: The method is new since MultiBoost adopts Q-ensemble.

-	Quality: For binary classification, the authors propose the complete optimization method. However, I doubt if it can generalize to the multi-class case. The algorithm also needs unlabeled data to estimate the conditional probability $Q(k|\cdot)$.

-	Clarity: The submission is clearly written except for the Section 3 Algorithm. Some errors occur in the description part of Section 3. For example, the definition of $F_k(\alpha)$ in line 130 should remove $\sum_{k=1}^p$. The equation below line 138 is a confusion notation for $F’_k(\alpha_{t-1}, e_{k,r})$.

-	Significance: The theoretical result is not novel and the guarantee can be anticipated.
For the multiple source dataset that the authors use to conduct experiments, some people propose the neural network-based methods, which can naturally cope with the multi-classification problem and have higher accuracy than boosting. Moreover, the authors should compare with other boosting methods to cope with multiple source problems, not just AdaBoost.

If the authors can use boosting to cope with multi-class classification and achieve high accuracy, I will consider changing my rate on the paper.

**Time Spent Reviewing:**

6

---

> ### Author Response · Authors · 2021-08-10
> **Author Response to Reviewer Nwre**
>
> We thank the reviewer for their comments and suggestions. Below, we address their main questions, in particular those relating to the multi-class extension of our algorithm.
>
> Generalization to the multi-class setting: our algorithm does admit a natural extension to the multi-class setting. We will add a full description of the algorithm to the appendix but can give here a brief description that should help convince the reviewer.
>
> In the multi-class setting, we have base classifiers $h$ mapping from $X \times Y$ to real numbers. Thus, $h(x, y)$ is the score given by $h$ to the class $y$ for the input $x$. There are different ways of tackling the multi-class setting in boosting. For simplicity, we only describe an extension similar to AdaBoost.MR, whose objective function is based on a sum over classes, and which is the most competitive multi-class boosting algorithm in practice (our algorithm admits all other natural boosting multiclass extensions). The form of the solution is then function $f$ defined for all $(x, y)$ by:
>
> $f(x, y) = \sum_{l = 1}^p \sum_{j = 1}^{N_l} \alpha_{l, j} \mathsf Q(l | x) h_{l, j} (x, y)$,
>
> and the objective function of our algorithm, MultiClassMultiBoost, is then the following:
> $F(\alpha)  = \max_{\lambda \in \Lambda} \sum_{k = 1}^{p} \frac{\lambda_k}{m_k}  \sum_{i = 1}^{m_k} \sum_{y \in \mathcal{Y}} \Phi \left( - \sum_{l = 1}^p \sum_{j = 1}^{N_l} \alpha_{l, j} \mathsf Q(l | x_{k, i})  \Big[ h_{l, j} (x_{k, i}, y_{k, i}) - h_{l, j} (x_{k, i}, y) \Big] \right)$.
>
> Line 129-130: that is indeed a typo, it should read $F(\alpha) = \max_{i = 1}^k F_k(\alpha)$. We will fix this and other typos.
>
> Theoretical results: our theoretical results are for a new multiple-source boosting objective with a corresponding new weak learning assumption and with Q-weighted ensembles. These results are all novel and include new and finer margin-based learning bounds for this problem. We are also presenting a key extension of our results and solutions to the Federated Learning scenario.
>
> MultiClass experimental results: we have fully implemented the algorithm during the rebuttal period to address the reviewer’s question and are reporting here the result of experiments using our MultiClassMultiBoost algorithm in the Fashion-MNIST dataset with 10 classes and with random forests used as base classifiers. We defined 3 domains: the first domain (k = 1) coincides with the original Fashion-MNIST; the second and third domains (k = 2, 3) are both defined as Fashion-MNIST with additive noise, with a different type of noise for k = 2 and k = 3. The errors reported below are multi-class classification errors for each source domain, k = 1, 2, 3, with RandomForest of depth up to 10 used as base predictors.
>
> AdaBoost.MR-1: [Error-1: 0.131, Error-2: 0.185, Error-3: 0.193]
>
> AdaBoost.MR-2: [Error-1: 0.157, Error-2: 0.152, Error-3: 0.159]
>
> AdaBoost.MR-3: [Error-1: 0.158, Error-2: 0.173, Error-3: 0.155]
>
> AdaBoost.MR-all: [Error-1: 0.141, Error-2: 0.166, Error-3: 0.163]
>
> MultiClassMultiBoost: [Error-1: 0132 , Error-2: 0.161, Error-3: 0.155]
>
>
>
> The results show that the multi-class version of our algorithm, MultiClassMultiBoost, significantly outperforms the AdaBoost.MR-k baselines (trained on single sources, k = 1, 2, 3) and the AdaBoost.MR-all baseline (trained on all sources combined). We will also report experimental results with neural networks used as base classifiers. Note, however, that for this task, random forests perform better than vanilla ReLU networks (see [1]). Overall, this demonstrates that our algorithm can achieve high accuracy in the multi-class setting in the challenging multiple-source problem with no pre-specified target domain.
>
> We will report several other multi-class experimental results in the final version.
>
> We are not aware of any other existing boosting algorithm dealing with the multiple-source problem we are considering, which does not assume a pre-specified target domain. However, in the final version, we will report comparisons with algorithms designed for a fixed target distribution, as suggested by another reviewer, and any other one suggested by this reviewer.
>
>
> [1] Han Xiao, Kashif Rasul, and Roland Vollgraf, “Fashion-MNIST: a Novel Image Dataset for Benchmarking Machine Learning Algorithms”, arXiv: 1708.07747, 2017.

---

### Official Review · Reviewer_shXz · 2021-07-16

**Rating:** 7
**Confidence:** 3

**Summary:**

The paper addresses the question of learning an accurate classifier using ensemble methods and based on multiple sources. The purpose of the classifier is to achieve good prediction on a set of possible future domains, modelled as linear combinations of the source domains. This is done by minimizing an agnostic loss function over the weights of a linear combination of source predictors, such as traditionally done in ensemble methods. The paper proposes a main algorithm solving this problem, as well as a version for the federated setting. A theoretical analysis and experimental validation are also proposed.

**Ethical Concerns:**

No ethical concern.

**Limitations And Societal Impact:**

The main limitation of the paper is, as said earlier, the strong role played by Q which is not discussed much.
The paper being more about solving a general task rather than focusing on specific applied problems, there is no societal impact in my opinion, neither positive nor negative.

**Main Review:**

Originality: Although the overall idea takes inspiration from various existing lines of work, the idea of building an ensemble predictor from multiple sources is to me novel. It builds upon two main well-known results: (1) the theory of ensemble learning, which states that a combination of weak classifiers can be a strong classifier; (2) transfer learning and domain adaptation, in which it is classical to weight the decision by the probability of the sources. The state of the art is briefly introduced in the main paper, but a bit more elaborated in the appendix. Maybe I would encourage the authors to insist a bit more on the transfer learning aspect which is only briefly discussed in the main paper.

Quality: The paper is technically correct, even though I have some reservations about the use of the term “boosting”. The main argument is based upon the result that AdaBoost corresponds to a greedy minimization of an exponential loss function. Following this idea, the authors build their own method around the minimization of a loss (which can be exponential, but not only) and minimization of this objective by application of coordinate descent. This is completely correct, but I wonder whether it is possible to really call that boosting. In its definition, boosting is meant to proceed to a sequential update of the predictor by sequential reweighting of the data according to the misclassification of previous models. It also builds upon the idea of applying a learning algorithm producing weak classifiers on several successive datasets (actually a same reweighted dataset). These aspects are of course present in the paper, but completely hidden in a very complex paragraph in Section 3. This comment is not fundamental and does not affect the overall quality of the paper, but I think a stronger discussion is needed here, and maybe I suggest that the authors weaken the importance of the word “boosting” in the paper. Also, I think that a short explanation about the connection between boosting and coordinate descent over exponential loss would be a very good thing to add, to be more accessible to a general ML audience which may ignore this connection. (While re-reading my review, I realize this comment is maybe more to be read from the “clarity” point of view than from the “quality” angle, but I still think that making the effort to explicitly show that the method is actual boosting would improve the quality of the paper).
The theoretical results seem correct to me, even though I have not reviewed (yet) the full proof of the two theorems.
Regarding the experimental validation, it is, in my opinion, sufficient to validate the interest of the method. However, I think that it may have been even more interesting to present results relative to other transfer learning / domain adaptation techniques which do not rely on boosting. In essence, the addressed problem is a transfer learning question and boosting is only a tool to solve this problem. I would find it more relevant to compare the results to TL algorithms which focus on the transfer of knowledge from the sources to the target, rather than to boosting methods which are designed to be single-task.
A point which would benefit from further discussion is the choice of the distribution Q(l|x). It is shortly mentioned that its choice is not direct, but I would have expected maybe more comments on that (which would be required for instance for a complete reproducibility). In the theoretical results, I am a bit surprised that the impact of this term is not visible in Theorems 1 and 2. After reading propositions 1 and 2, it was clear that this term was important for the quality of the predictions, and yet it seems not to play any role in the margin bounds. Could the authors comment on that point? Finally, if not done in theory (which I understand would be difficult – if even possible), I regret that the influence of this term is not highlighted in the empirical section, pointing out how a wrong choice of Q could affect the overall algorithm.

Clarity: The paper is rather well-written, even though the notations may be sometimes difficult to follow. Unfortunately, this is mostly due to the addressed problem itself and I do not have any real correction suggestions for that. Maybe section 2 could be slightly corrected to introduce the notations in a more readable way? This is only a detail, but I think it would significantly help the reader. During my reading, I had to go back to the notations multiple times and had some difficulties retrieving them.
In terms of introduction of the ideas, I particularly enjoyed the first two propositions which are a smart way to introduce the Q(l|x) term. Minor comment, though: “probability of a domain given an input point” seems quite ambiguous. I suggest it to be replaced by a clearer description, for instance “probability that an input point belongs to a domain”.

Significance: The results are somehow significant. The main scenario is probably the weakest in terms of its induced advances in the state of the art, but the federated learning version is extremely important in my opinion and offers a clean and well-justified solution to a very important problem.


**Time Spent Reviewing:**

3-4

---

> ### Author Response · Authors · 2021-08-10
> **Author Response to Reviewer sHxz**
>
> We thank the reviewer for their careful reading and appreciation of our work and for their constructive suggestions. Below, we respond to the main comments.
>
> - Transfer learning literature: we will insist more on that in the main text, as suggested.
>
> - Paragraphs in section 3: we will give a stronger discussion for the paragraphs in section 3, as suggested by the reviewer, as well as for the relationship between boosting and coordinate descent.
>
> - Experiments with transfer learning algorithms: we will include comparisons with other multiple-source algorithms and a fixed target distribution, as also suggested by another reviewer.
>
> - We will elaborate more on the choice of Q and its training. In our experiments, we used multinomial logistic regression to model Q.
>
> - Role of Q in theoretical results: the reviewer is correct, the choice of Q as a domain classifier is important. We have given a brief analysis pointing out that importance in lines 161-168. We will further elaborate on that in our theory section. In short, the proper choice of Q as a domain classifier leads to more favorable empirical margin losses, while it does not influence the complexity terms present in the bound
>
> - Role of Q in empirical results: In Figure 3, we have sought to illustrate the importance of the conditional probabilities. But, the reviewer's suggestion is very good, we will seek to further show empirically that the definition of Q as a domain classifier is important by reporting results with a poor or ‘perturbed domain classifier’.
>
> - Notation: we will seek to improve that and make it more accessible, although, as indicated by the reviewer, this is partly due to the inherent presence of multiple sources.

---

### Official Review · Reviewer_2m7P · 2021-07-19

**Rating:** 7
**Confidence:** 4

**Summary:**

This paper considers the domain adaptation setting where we are given training examples from $k$ different distributions $D_1, \cdots, D_k$. We assume a hypothesis class $H_k$ the at is good for each domain (in the weak learning sense). At test time, the samples are drawn from a convex combination $D = \sum_i \lambda_i D_i$. The goal is to find one linear combination of  weak learners which will perform well for any convex combination. It turns out that this is not possible, but that if you allow the weights to depend on the conditional probability of coming from a particular domain $D(j|x)$, then a single learner is possible. This I believe is already proved in the paper of Mansour-Mohri-Rostemizadeh 2008, although the result there relies on a fixed-point theorem and perhaps does not give an efficient algorithm.

The main contribution of this work is a boosting style algorithm that achieves this  guarantee. The algorithm does simple co-ordinate wise gradient descent like in Adaboost. They show theoretical margin-based guarantees for their algorithm, and experimental results showing that it outperforms more naive boosting-based approaches.

**Limitations And Societal Impact:**

Yes

**Main Review:**

I am a little concerned about the relation between this paper, and the work of Mansour-Mohri-Rostemizadeh 2008 [MMR]. At the very least, I have problems with how this paper credits that paper without adequate citations. It is possible that there is some substantial overlap in terms of ideas.

1. The notion of $Q$-ensembles:
This paper says: *Instead, we put forward Q-ensembles, which are convex combinations weighted by a domain classifier Q, that is, Q(k|x) is the conditional probability of domain k given input point x. This crucially differentiates our work from that of past studies*

I am not sure I agree with this statement. It appears that the notion of $Q$-predictors is present (or at the very least implicit) in [MMR]). In Section 5.2, the predictor $h_z^0$ they use is exactly the $Q$-weighted predictor where the conditionals are computed for the convex combination of distributions with coefficient $z$).
In that paper the setting is that there is a single hypothesis $h_l$ for each $l \in [k]$, and we do not get access to samples. It seems that if one considers $h_l = \sum_j \alpha_{l, j}h_{l,j}$ as a single predictor that does well for domain $D_l$, then you get exactly the form of predictors considered in this paper:
$$ f(x) = \sum_{l =1}^pQ(l|x)\sum_{j \=1}^{N_l}\alpha_{l, j}h_{l, j}(x) $$
In particular, this paper never spells out explicitly whether these conditionals are being computed with regard to the uniform convex combination of some other combination, it would be nice to clarify this.

2. Existence of good $Q$-ensembles:
Beyond the notion of $Q$- suggests that the existence of the kind of predictor that this paper gives is implied [MMR]. It seems that their linear combinations are not explicit owing to a fixed point argument, whereas this paper gives an explicit algorithm using boosting.

3. The negative results:
Proposition 1 in this paper seems identical to Theorem 1 in MMR08, similarly I believe Proposition 2 is just Theorem 2. But there is no citation or acknowledgement, and a reader might think this is a new contribution. While this was probably just oversight, it is not good writing.

While I like aspects of this paper, I am going with a weak reject for now based on these issues. This is only meant as a placeholder. I would be open to revising my scores if the authors can give a rebuttal that:
- Accurately acknowledges what is derived (directly or easily) from prior work.
- Conditioned on this, emphasizes what is truly novel here.


----------------------------------------------------------------------------------------------------------

After reading the responses, I am convinced that this paper makes a substantial original contribution. I am raising my score to accept.

**Time Spent Reviewing:**

3.5

---

> ### Author Response · Authors · 2021-08-10
> **Author response to Reviewer 2m7P**
>
> We thank the reviewer for their comments. In short, the reviewer is correct, we should and will credit the paper of [MMR] better and we will also clarify the relationship with our work. Rest assured that this was indeed just an oversight as further explained below. Here are some more detailed responses.
>
> Q-ensembles and connection with paper of [MMR]: we are referencing that paper and the related work of Hoffman et al. We agree with the reviewer, however, that we should add more citations to this work in several paragraphs and will definitely do so. This was not intended as a poor credit assignment but rather a poor decision at some stage to move the discussion of some related work to the appendix (due to lack of space) and also a decision to limit repeating the same reference. Our statement about ‘previous work’ in the context mentioned by the reviewer only refers to previous work on boosting or ensemble methods. We will clarify that and give better references and credit to that work.
>
> Let us emphasize, however, that the setting studied by [MMR] is different from ours: it is one where the learner does not have access to labeled data and where, instead, an accurate predictor is available for each domain. The solution proposed by [MMR] consists of a distribution-weighted combination, based on estimates of the source distributions (density estimation). In contrast, in our setup, the learner does have access to labeled data and does not have access to accurate pre-trained predictors for each domain and our Q-ensembles are based on conditional probabilities of each domain, not density estimates. Furthermore, the combination rules considered in [MMR] are normalized while our Q-ensembles need not be, which is key in boosting in general. Thus, the forms of the solutions are different.
>
> Q conditionals: our conditional probabilities Q are not defined ‘with regard to the uniform convex combination of some other combination’. They are simply derived by estimating the probability of each domain (lines 304-306), using multinomial logistic regression (or conditional maximum entropy).
>
> Existence of good Q-ensembles: as already mentioned, the setting studied by the paper [MMR] is different. The existence of a ‘good predictor’ in that work refers to a good combination of existing accurate pre-trained models, one for each domain. Indeed, in our context, we do not have access to already existing good predictors and instead are providing an explicit boosting algorithm making use of *all* the labeled training data to come up with a single good predictor overall, without seeking an accurate per domain model. We will report in our final version a comparison with the MMR solution used with source models obtained via single-source AdaBoost.
>
> Proposition 1: this is indeed similar to theorem 1 in [MMR]. We will definitely reference that work and indicate the connection. Let us add, however, that although the result is similar, there are some technical differences. In particular, we are considering here a binary classification loss, while the result of [MMR] is for the absolute loss.
>
> Novelty: to summarize, the idea of Q-ensembles in our boosting context is inspired by the distribution-weighted combinations of [MMR] and the basic motivation via proposition 1 and its proof are similar. However, the main technical content of this work and the contributions are all novel.  Our formulation of the multiple-source boosting problem, including our weak learning assumption, our algorithmic solutions, including an extension to the Federated Learning setting, and our theoretical analysis of the problem (Lemma 1, Theorems 1, and 2), including finer margin-based learning bounds, our extensive experimental results, are all novel and unrelated to [MMR].

---

> > ### Comment · Reviewer_2m7P · 2021-08-24
> > **Question about comparison to MMR**
> >
> > Thanks for your response. While I see the differences, I am still not clear about the answer to the following question: suppose use the samples from each domain to run Adaboost on that distribution, and then combine those predictors using the scheme suggested by MMR. How would that algorithm compare to your algorithm theoretically? What guarantees does your algorithm give which that predictor would not have? Conversely, are there guarantees that MMR gives which are stronger than what you get?

---

> > > ### Author Response · Authors · 2021-08-26
> > > **Response to Reviewer 2m7P: Comparison to MMR**
> > >
> > > We appreciate the reviewer's question and his understanding of the key differences we pointed out.
> > >
> > > The predictor for the algorithm described by the reviewer in general does not benefit from informative guarantees using [MMR], as discussed below.
> > >
> > > 1. target labeling function assumption: the main analysis and results of [MMR] require that the target labeling function (or conditional probability of Y given X for an extension of that analysis) be the same for all domains. This is a strong condition that may not hold in practice and that is not required for our learning guarantees for MultiBoost.
> > >
> > > 2. loss function: the guarantees for [MMR] hold only for a continuous loss function, since they rely on Brouwer's fixed-point theorem. In particular, they do not hold for the binary or multi-class mis-classification losses considered here.
> > >
> > > One can resort instead to a convex surrogate loss (such a guarantee would be then in terms of the convex loss of the predictors to combine and not their more favorable zero-one loss). But the [MMR] guarantees also require the loss to be bounded, which would not hold for an unbounded domain. Even for a bounded domain, the bound could be large and the value of the convex loss on the boundary even larger (exponentially larger for AdaBoost) thereby making the desired guarantee essentially vacuous. In contrast, our guarantees hold for the zero-one loss.
> > >
> > > 3. Algorithms: the technique of [MMR] requires density estimation for each domain. With estimated densities, their guarantee becomes somewhat looser. Furthermore, the solution is not based on a convex optimization and thus cannot directly benefit from the theoretical bound, even if it could be applicable (see above).
> > >
> > > 4. hypothesis set: ignoring the normalization, which does not affect the definition of the classifier, the [MMR] solution suggested by the reviewer is a specific element of the family of ensembles our algorithm and learning bounds consider. Our algorithm seeks the most favorable ensemble using all the available labeled data, which in general could be quite different and more favorable than the  predictor obtained by combining MMR and Adaboost.
> > >
> > > More generally, our algorithm is not restricted to deriving intermediate predictors for each domain and instead directly exploits the labeled training samples from all the sources simultaneously to find a single good predictor.
> > >
> > > Consider the extreme case where all sources follow the same distribution and all training sets have the same size. The algorithm suggested by the reviewer would first train one AdaBoost model $f_k$ for each source. Assuming perfect density estimation, [MMR] would then return the uniform average $\frac{1}{p} \sum_{k = 1}^p f_k$ with each $f_k$ trained on $m/p$ samples. Instead, MultiBoost returns a single model trained on all $m$ samples that in general can be far superior.
> > >
> > > To further illustrate this, we ran this experiment on the SVHN dataset with 3 ‘sources’ defined by sub-samples from an original training sample. This led to an error of $26.1$% for the predictor obtained via [MMR] and only $22.8$% for MultiBoost.  We will include more experimental comparisons with that algorithm in the final version.

---

### Official Review · Reviewer_8DgA · 2021-07-27

**Rating:** 7
**Confidence:** 4

**Summary:**

Utilizing recent formulations/results, the authors propose MutliBoost, a multi-source ensemble learning algorithm where: (1) it is assumed that any target domain is a mixture of the source domain distributions and (2) the learned function form is a convex combination of base functions weighted by a domain identification classifier, Q(k|x), and referred to as Q-ensembles. The domain identification classifier can be learned from unlabeled data (other than the domain label) and an important innovation as it leads to a more natural multi-source Boosting objective and generalizes many existing (and widely-used) multi-source Boosting works. This formulation aligns with recent results regarding agnostic loss [Mohri, Siven, and Suresh; Agnostic Federated Learning, ICML19] and is able to use related findings to prove favorable theoretical properties wrt previous work and leads to FedMultiBoost, a federated learning variant. Experiments are conducted on five binary datasets converted to multi-domain settings (sentiment analysis, hand-writing recognition, object recognition, and UCI adult tabular data) against natural single domain transfer and combined domain data baselines, showing statistically significant improvements. Several additional experiments are included in the Appendices to also show further results (e.g., domain-identification classifier, FedMultiBoost) that addresses multiple natural questions in the paper.


**Limitations And Societal Impact:**

== Technical Limitations ==
While the primary limitations are stated, there isn’t an explicit section and/or discussion of the ramifications of the relatively strong assumptions. Additionally, as stated, the empirical results don’t have a notable discussion section.

== Potentially Negative Societal Impact ==
No concerns in this regard — it is an ML algorithms paper with a theoretical focus. Nothing that isn’t beyond standard ML concerns.

**Main Review:**

As previously stated, this submission primarily builds on recent multi-domain and federated learning finding to revisit and reformulate a multi-domain Boosting algorithm with improved properties relative to existing formulations. Given that this is a well-visited area, the formulation is notable in that it does provide practical advantages (training for all target mixtures) and preferable theoretical properties. Additionally, it is an elegant formulation and has potential to restart work in an area that has become somewhat saturated. The theoretical analysis is sound, concise, and sufficiently thorough. The empirical results are promising, but would benefit from additional comparisons to existing work and, if possible, a ‘real-world’ (non-benchmark) dataset to demonstrate practical strengths of (Fed)MultiBoost. Below, I will evaluate the work along the requested dimensions, but overall think that this is a worthwhile contribution with sufficient originality, quality, clarity, and potential significance (with strengths in theoretical analysis and presentation outweighing weaknesses in empirical evaluations) and lean toward accepting.

== Originality ==
The multi-source Boosting problem is not new, but this formulation provides a generalization to existing approaches that is theoretically more broadly applicable (given the weak learning assumption can be satisfied, the domain-identification classifier is sufficiently accurate, and sufficient training data exists). Additionally, the analytical methods are primarily derived from existing work — so this isn’t the strength of originality either. Specifically, core findings are largely derived from earlier margin-based analysis of Boosting or agnostic learning results in federated learning scenarios (i.e., this is not an analytical breakthrough). However, while I think many people could have put these ideas together if I were to say “adapt [Mohri, Siven, and Suresh; Agnostic Federated Learning, ICML19] to create a multi-source Boosting algorithm”, I don’t think making this connection was obvious nor was the analysis trivial.

== Quality ==
The scholarship and theoretical analysis is solid. The paper is well-contextualized wrt the first generation of Boosting work, application of this work to domain adaptation/multi-source learning, and application of recent work in multi-source/federated learning. The development of the theory is self-contained and answers most of the relevant questions. From a practical perspective, the underlying assumptions in [lines 101-108] seem potentially overly strong as a weak learner must exist for any target-mixture — which may limit it’s practical application. While the empirical results are good in terms of what is presented, they also don’t establish that this is a clearly practical algorithm since: (1) it is not clear why there aren’t results comparing to [Yao & Doretto, 2010] for the single-target case as it would demonstrate the sensitivity of accurate domain identification to some degree and (2) there is no (non-benchmark) ‘real-world’ dataset to prove it is clearly practical. AdaBoost-all is a natural baseline, but it would be surprising if MutliBoost didn’t perform better. Additionally, while space is a problem, but additional discussion of the experimental results would be useful as it is presently a “we outperform fairly weak baselines”. This is somewhat mitigated by the appendices that add to the results and are of quality similar to the body of the main paper. Also, FedMultiBoost is an extension of this connection additional considerations with softmax explored — which may influence other related works.

== Clarity ==
Overall, the paper is well-developed and clear. While reading the most directly related works does help with understanding, a famliar reader will likely view the paper as self-contained and well-focused on the most important new results. Again, the appendices add to the paper (without being strictly necessary) although I think there are opportunities to better reference them with short summary statements in context. Also, from a scholarship perspective, better references in Sections 3 (to older Boosting work, etc.) would be helpful and maybe a bit more forward/in context in section 4 (although lines 218-223 are sufficient). However, I believe clarity is a strength of this work.

== Significance ==
Multi-source is popular and the tie to federated learning increases the potential for impact; I would expect this to be the preferred starting point for multi-source Boosting approaches. The primary limitation that potentially impedes significant is that it isn’t clear that it is the SoTA since thorough empirical comparisons to existing multi-source settings or a ‘real-world’ setting. From a theoretical perspective, the analysis draws heavily from existing work, but after (re-)reading those papers, I liked this presentation and I believe will influence other analyses

**Time Spent Reviewing:**

4.5 hours

---

> ### Author Response · Authors · 2021-08-10
> **Author response to Reviewer 8DgA**
>
> We thank the reviewer for their careful reading and detailed comments. Below, we provide some responses and clarifications.
>
> Originality: the reviewer already knows that, given their careful reading, but we wish to emphasize that our formulation of the multiple-source boosting problem, our algorithmic solutions, including an extension to the Federated Learning setting, and our theoretical analysis of the problem, including finer margin-based learning bounds, are all novel.
>
> Assumption: our weak learning assumption (lines 101-108) is in fact only for each source domain and not for target mixtures. Thus, we view this as an assumption that is not stronger than the standard weak learning assumption for a single domain.
>
> Experimental results: we are already presenting an extensive set of experimental results with natural baselines and multiple datasets. We are not aware of any existing algorithm dealing with the boosting scenario we study. But, we are happy to include comparisons with other algorithms such as ​​[Yao & Doretto, 2010], which are designed for a specific and pre-given target distribution, as a point of comparison, or any other algorithm that the reviewer recommends.
>
> Our algorithm is very practical and its running time is similar to that of standard AdaBoost. That MultiBoost significantly outperforms AdaBoost-all is indeed an important benefit of the algorithm that we are demonstrating empirically. AdaBoost-all is in fact the algorithm used in this context by many in practice. We have used several datasets commonly used in previous work dealing with multiple sources. We would be happy to report results with other publicly available datasets the reviewer would suggest that naturally admit multiple sources.
>
> Significance: we have sought to present empirical comparisons with natural baselines since we are not aware of any boosting algorithm for the scenario we study. In that sense, our algorithm can be viewed as providing the state-of-the-art: it is outperforming natural boosting baselines and benefits from strong guarantees. However, as already mentioned, we are happy to report comparisons with other algorithms the reviewer would suggest. Our FedMultiBoost algorithm is also likely to be a very significant contribution in Federated Learning.

---

> > ### Comment · Reviewer_8DgA · 2021-09-01
> > **thanks for the detailed comments**
> >
> > After reading the other reviews and the rebuttal, I am more confident that this is a good paper. Thanks.

---

### Decision · Program_Chairs · 2021-09-27

**Decision:**

Accept (Poster)

**Comment:**

This paper introduces and analyzes a new boosting algorithm for learning in the presence of multiple source domains. We were persuaded by the author response to the concerns expressed in the reviews, especially about novelty wrt MMR.